# GeoDE: a Geographically Diverse Evaluation Dataset for Object Recognition

**Vikram V. Ramaswamy**[1], **Sing Yu Lin**[1], **Dora Zhao**[2]*, **Aaron B. Adcock**[3],
**Laurens van der Maaten**[3], **Deepti Ghadiyaram**[4], **Olga Russakovsky**[1]

[1]Princeton University    [2]Stanford University    [3]Meta AI    [4]Runway
*Work done as a student at Princeton University

## Abstract

Current dataset collection methods typically scrape large amounts of data from the web. While this technique is extremely scalable, data collected in this way tends to reinforce stereotypical biases, can contain personally identifiable information, and typically originates from Europe and North America. In this work, we rethink the dataset collection paradigm and introduce GeoDE , a geographically diverse dataset with 61,940 images from 40 classes and 6 world regions, with no personally identifiable information, collected by soliciting images from people around the world. We analyse GeoDE to understand differences in images collected in this manner compared to web-scraping. We demonstrate its use as both an evaluation and training dataset, allowing us to highlight and begin to mitigate the shortcomings in current models, despite GeoDE's relatively small size. We release the full dataset and code at `https://geodiverse-data-collection.cs.princeton.edu/`

## 1    Introduction

The creation of large-scale image datasets has enabled advances in the performance of computer vision models. Previously limited by internal manual collection efforts [13, 15, 12], in the past 15 years the size of these datasets has rapidly grown. This growth has been empowered by a new data collection framework: scraping web images at scale. These images are either human-labelled (e.g., ImageNet [8, 26]), use tags (e.g., CLIP-400M [24]) or use self-supervision (e.g., PASS [2]).

However, these web-scraped datasets come with their downsides. One of these downsides is that these datasets can often contain pernicious social and cultural biases. For example, gender and racial biases can manifest through underrepresentation and/or through stereotypical depictions of certain demographic groups [21, 4, 40, 34, 3]. There is also *geographic bias*: works of e.g., Shankar et al. [27] and de Vries et al. [7] show that web-scraped datasets consist of images mostly from North America and Western Europe.

The other common downsides are copyright, consent and compensation. Dataset creators frequently do not obtain full permission of the content creators and of the people featured in the content [3].[1] While annotators are compensated, content creators and image subjects rarely are [3]. Though there have been efforts to balance datasets [10], clean datasets [36], and protect privacy of depicted subjects by blurring [39], methods that rely on web-scraping cannot fully eliminate these issues [3, 18].

To tackle these issues, an exciting new dataset DollarStreet [25] was recently introduced (licence: CC). Instead of web-scraping, DollarStreet sources data from the Gapminder foundation. It comprises of images taken by volunteer and professional photographers in different countries to illustrate households with different income statuses. This results in 38,479 images from 63 different countries, tagged with 289 labels. DollarStreet overcomes issues of consent, and is in many ways the first truly geographically diverse dataset (Tab. 1).

---

[1]While images used are sometimes under the most permissive Creative Commons license, it is unclear if creators know the full impacts of their images being used in the training of large scale models.

37th Conference on Neural Information Processing Systems (NeurIPS 2023) Track on Datasets and Benchmarks.

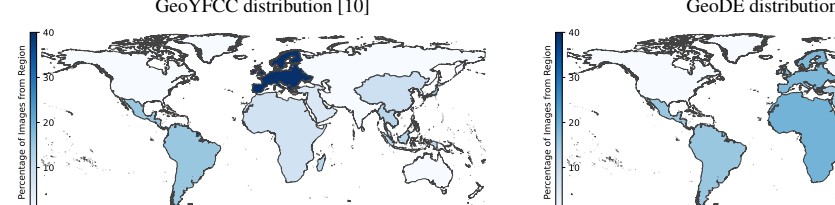

Figure 1: We construct a geographically diverse dataset GeoDE that is approximately balanced across 6 world regions. We visualize the images per region, and compare our distribution (*right*) to that of a previously created diverse dataset GeoYFCC [10] (*left*).

However, DollarStreet curates a computer vision dataset from images that have *previously been collected* and released on the web. In contrast, we present an alternative geographically-diverse data collection process (more details in Sec. 3), which allows us to explicitly target a different object-centric image distribution.

Further, with the advent of multimodal foundation models [24, 22, 23, 32, 9], revising the idea of manual collection of test datasets may be in order. With these models, direct access to the training data is frequently impossible, although we know they were trained on trillions of web-scraped examples. Thus, any dataset comprising of previously available images may actually have been included in the model training, violating the core machine learning tenet of separate train/test splits.

**Contributions.** We collected a **geographically diverse dataset of common objects** which overcomes many issues described above. Concretely, we *commission* photos of different objects from people across the world from Appen (`www.appen.com`)'s global workforce. This naturally resolves **consent** concerns, similar to DollarStreet [25] and ensures that the data is (at least temporarily) an **unseen test set**. We **own the copyright** to all of the images in GeoDE, **have explicit permission** from creators to use these images for computer vision, verified that the images **do not show identifiable people** or other personally identifiable information (PII), and ensured **compensation** to the content creators.

The resulting Geographically Diverse Evaluation (GeoDE) dataset[2] contains 61,940 images roughly balanced across 40 object categories and 6 geographic regions. We find that the object recognition problem becomes surprisingly challenging since the GeoDE images represent the **diverse appearance** of common objects across six global regions: Africa, the Americas, East Asia, Europe, Southeast Asia, and West Asia. Similar to de Vries et al. [7], we show that modern object recognition models perform poorly on recognizing objects from Africa, East Asia, and SE Asia. Augmenting current training datasets (like ImageNet [8, 26]) with training images from GeoDE yields an improvement of 9% on DollarStreet [25] and 21% on the test split of GeoDE .

Further, requesting images with specific content **mitigated selection bias**: web-scraped images are typically uploaded by creators with different incentives, e.g. to generate engagement with exciting/unique content [28], and disincentives for mundane everyday content. We show that the distribution of images in GeoDE is different to that in other datasets, even when controlling for world region and object class.

Unfortunately, a key drawback of our method is the ***cost***, which is partially the result of aiming for fair compensation to the content creators and curators. The method can also lead to other biases (e.g., lack of economic diversity, since workers are required to have a smart phone). However, we demonstrate that even small amounts of data collected in this way can be beneficial in remedying some of the concerns with large-scale web-scraped datasets. Our work challenges the current paradigm for dataset collection and illustrates the process of manually curating image datasets. In addition, we introduce the GeoDE  dataset which offers a more geodiverse and ethically created alternative for object recognition. Data and code can be found at `https://geodiverse-data-collection.cs.princeton.edu/`

## 2   Related Work

There are three key research directions that inspired this work. The first is the call to increase geographic diversity in visual datasets [27, 7]. In response, there have been attempts to construct such datasets [10, 1], summarized in Tab. 1. However, these datasets are still geographically concentrated (in Europe for GeoYFCC [10] or India for OpenImages Extended [1]). The dataset most similar to

---

[2]GeoDE is owned and maintained by Princeton; Meta AI was not involved with the data collection.

Table 1: We compare approaches to dataset collection, along with the distribution and sizes of each. Although GeoDE is smaller than standard datasets, we ensure that images are sourced with permission, contain no identifiable people, and are balanced across both regions and object classes.

| Dataset | Size; distribution | Collection process; annotation process | Geographic coverage | Personally Identifiable Info (PII) |
|---|---|---|---|---|
| ImageNet [8, 26] | 14.2M; mostly balanced across classes | Scraped images from the web based on the class label; crowd-sourced annotations | Mostly North America & Western Europe [27] | Contains people, subset with faces blurred [38] |
| PASS [2] | 1.4M; N/A (no labels) | Random images from Flickr; no annotations | Flickr, thus mostly North America & Western Europe | No people |
| OpenImages [19] | 9.1M ; long-tailed class distribution | Flickr images with CC-BY licenses; automatic labels with some human verification | Mostly North America & Western Europe [27] | Contains people |
| OpenImages Extended [1] | 478K; long-tailed class distribution | Crowd-sourced gamified app to collect images; automatic labels and manual descriptions | More than 80% of images are from India | People are blurred |
| GeoYFCC [10] | 330K; long-tailed class distribution | Flickr images subsampled to be geodiverse; noisy tags | 62 countries, but concentrated in Europe (Fig. 1) | Contains people |
| DollarStreet [25] | 38,479; mostly balanced across topics | Images by professional and volunteer photographers; manual labels including household income | 63 countries in Africa, America, Asia & Europe | Yes, with permission |
| GeoDE (ours) | 61,940; balanced across classes and regions | Crowd-sourced collection using paid workers; manual annotation | Even coverage over six geographical regions (Tab. 3) | No identifiable people and no other PII |

ours is **DollarStreet** [25], also mentioned in the introduction. Both DollarStreet and our GeoDe were collected to improve geographic diversity of image datasets, are relatively small scale datasets (62k for GeoDE, 39k for DollarStreet), and are collected through crowd-sourcing. However, DollarStreet was repurposed as a computer vision dataset by curating the images through GapMinder, a non-profit organization that collected these images to showcase differences in how people live around the world. Thus, the images in DollarStreet were collected through a more social science perspective: for example, images are collected to showcase different everyday actions such as "washing hands" as opposed to objects such as "hand soap". On the other hand, GeoDE is more focused on understanding how objects across the world are visually different.

Second, in using participants to *generate* visual content, we follow video datasets Charades [28], Epic Kitchens [6] and Ego4d [14]. We similarly leverage paid workers to provide examples that fall outside the common web-scraped distribution. However, we differ in that our key goal is to ensure geographic diversity. This poses unique challenges in recruitment and dataset scope (more in Sec. 3).

Finally, in our data collection efforts we take into account the extensive literature around selection bias in computer vision datasets [31, 4, 40, 34, 29, 11, 37, 3] and ensure that our dataset is collected responsibly, with attention to privacy, consent, copyright and worker compensation [3].

## 3   Collecting GeoDE

We describe our data collection process, including our selection of object classes and world regions.

**Selecting the object classes.** We focus on object classes that are likely to be visually distinct in different parts of the world. Selecting such objects is a chicken-and-egg problem: without a geographically diverse dataset at our disposal, it is unclear which objects are diverse. We adopt a number of heuristics using existing datasets to find a plausible set. We use simple computer vision techniques (linear models and visual clustering, using features extracted from self-supervised PASS-pretrained models [2]) along with manual examination to identify a set of candidate tags from DollarStreet [25] and GeoYFCC [10] (e.g., "chili," "footstool," "stove"). To prune these tags, we remove those that are not objects (e.g., "arctic", "descent"), remove wild animals not found in all regions (e.g., "gnu", "camel") and group variants of objects (e.g., "stupa", "temple", "church" and "mosque"). The final set of objects is in Tab. 2, and the full process is in the supp. mat.

Table 2: GeoDE consists of 40 object classes, loosely organized into 4 groups.

| *Indoor common* | *Indoor rare* | *Outdoor common* | *Outdoor rare* |
| --- | --- | --- | --- |
| bag, chair, dustbin, hairbrush/comb, hand soap, hat, light fixture, light switch, toothbrush, toothpaste/toothpowder | candle, cleaning equipment, cooking pot, jug, lighter, medicine, plate of food, spices, stove, toy | backyard, car, fence, front door, house, road sign, streetlight/lantern, tree, truck, waste container | bicycle, boat, bus, dog, flag, monument, religious building, stall, storefront, wheelbarrow |

Table 3: GeoDE consists of images from six world regions. Within each, there are 3-4 countries contributing to most of the images. Participants from other countries within the region were accepted.

| | | | |
| --- | --- | --- | --- |
| *West Asia:* Saudi Arabia, UAE, Turkey | | *Africa:* | Egypt, Nigeria, South Africa |
| *East Asia:* China, Japan, South Korea | | *SE Asia:* | Indonesia, Philippines, Thailand |
| *Americas:* Argentina, Colombia, Mexico | | *Europe:* | Italy, Romania, Spain, United Kingdom |

**Selecting diverse geographic regions.** We chose six regions: Africa, Central and South America ("Americas"), East Asia, Europe, SouthEast ("SE") Asia and West Asia. Within each, we targeted 3-4 countries (Tab. 3). The regions were chosen due to the lack of available images from them in most public datasets [27, 7, 35]; the countries were chosen based on the presence of participants within Appen's database[3]. We obtain a roughly even distribution of images across each class and region pair.

**Image collection and worker demographics.** Workers were asked to upload images for a given object class (Fig 2). There were 4,500+ workers, representing a range of genders, ages and races (see supp. mat.). All images submitted were manually checked by Appen's quality assurance (QA) team.

## 4    Lessons learned from collecting GeoDE

A key part of this study was to understand if manually taking photographs is a viable alternative to web-scraping. In this section we detail the lessons learned in constructing the GeoDE dataset.

**Getting sufficient images of all object classes.** While some object classes expectedly proved more difficult than others (e.g., "monument" or "flag" were simply hard for workers to find), others surprised us. For example, "stove" was originally underrepresented until the definition was clarified to "any cooking surface either electric, gas, induction." Workers' perception of their cooking appliance as "stove" varied, highlighting a vocabulary challenge unique to geographically diverse data collection.

**Multiple copies of images.** Two most common types of error were incorrect images (i.e, not belonging to the class selected) and multiple copies of an image. The QA team found that participants often submitted multiple copies of the same object instance from different angles despite instructions not to do so. Workers also sometimes submitted very similar objects (examples in supp. mat.), for example, three hats by the same worker with slightly varying colors). We filtered out such images.

**Multiple objects per image.** For some of the (especially larger and outdoor) object categories, it was difficult to ensure that other objects were not present, particularly "trees". For example, we found that images of "fences" often have "trees" present, and it was not always possible to discern between objects in the foreground and background (examples in supp. mat.). Thus, we requested that images that had a significant portion of the image covered by trees be explicitly tagged. These additional annotations can be used to filter and remove such images and/or to analyze errors made by a model.

**Other.** Beyond these, the rest of the data collection went smoothly. Following instructions, only 0.78% images contained identifiable information. Some images contain non-identifiable incidental people in the background (especially for larger object classes, like "monument"). All such images are tagged in GeoDE. We were also able to ensure that the number of images per region is roughly equal, although it was harder to obtain an even number of images per country within each region.

**Cost.** Collecting images in this way was expensive: each image cost roughly $0.87 for a total cost of $54,000, not including researcher time. This allowed us to compensate photographers as well as the management and QA teams for their labour.

---

[3]While we do not currently find this to be the case empirically, we acknowledge that the regions themselves are quite broad, and certain objects might look different within the region.

---

*General Instructions*

In this task, you will submit up to **3 photos** of the **same type of object** (e.g., upload 3 photos of **3 different bags**; please **do not** upload 3 photos of the same bag from different angles).

1. Please make sure the location function is enabled for the camera.
2. The photo resolution should be at least 640 x 480.
3. All images should be new photos captured with Appen Mobile.
4. Please make sure it's a single object per image.
5. Please make sure it's a well-lit environment and the object is clearly visible in the photos.
6. Please make the object occupy at least 25% of the image.
7. Objects captured are foregrounded and not occluded.
8. Objects should not be blurred, e.g., motion blur.
9. No effects or filters added (cropping is acceptable).
10. Please try to avoid capturing people in the images (it's OK if people are blurry in the background and far from the camera).
11. Please try to avoid capturing vehicle license plates in images.

---

Figure 2: Image collection instructions given to workers.

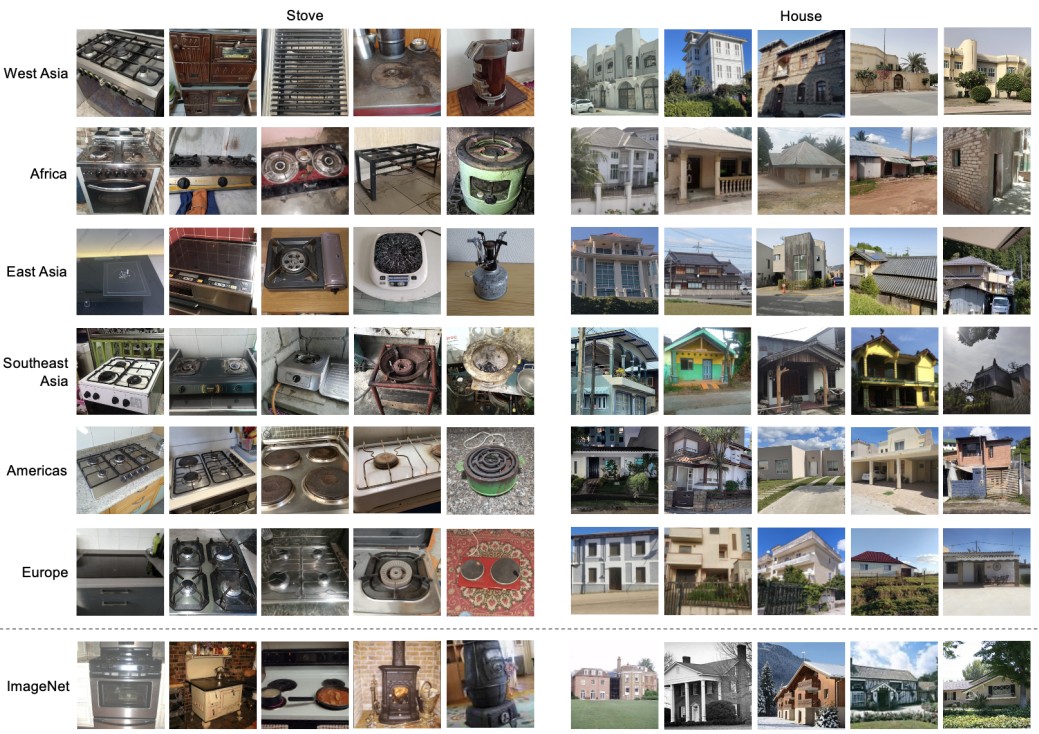

Figure 3: Sample images of two object classes in different regions within GeoDE (and ImageNet in the bottom row, for comparison). Note the variety of stoves and houses across geographic regions in GeoDE – and also the fact that the stoves are more *used* (thus arguably more realistic) than in ImageNet. Product labels on images in the figure have been blurred.

## 5 Comparing GeoDE to current datasets

We compare GeoDE with three datasets: the canonical ImageNet [8], the more geographically diverse (but still webscraped) GeoYFCC [10] and finally, the recently curated DollarStreet dataset [25].

**Qualitative.** In Fig. 3, we show a subset of 60 GeoDE images of "stoves" and "houses" (more in the appendix). Compared to images from ImageNet, we see a larger variety of stoves: e.g., induction coils, single and two burner stoves. The stoves also appear more *used* than those in ImageNet. Similarly, for "house," we see a larger range in terms of materials used and size. In the Secs. 6 and 7 we examine the impact of this diversity on visual recognition models.

**Statistics.** GeoYFCC [10] (license: CC) is a webscraped dataset subsampled from YFCC100M [30] to be geographically diverse; thus, by raw counts it is a much larger dataset compared to both DollarStreet and GeoDE, with over 1M images from 62 countries. However, looking at the regions

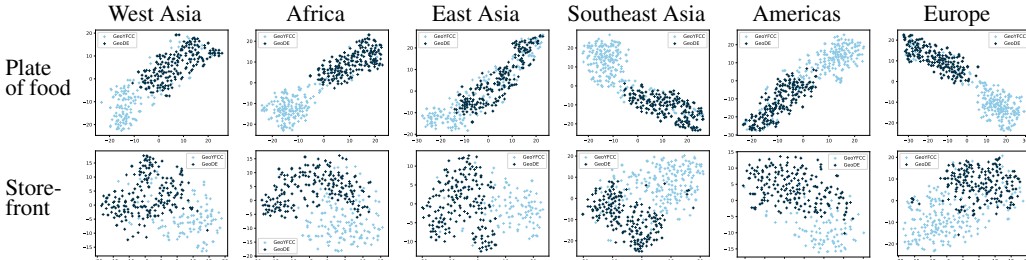

Figure 4: We visualize the TSNE plots for several of object classes per region for GeoYFCC (light blue) and GeoDE (dark blue). While the features do overlap slightly, on the whole, they are very different for dataset distributions, even within each (region, object) tuple. (See Fig. 5 for DollarStreet)

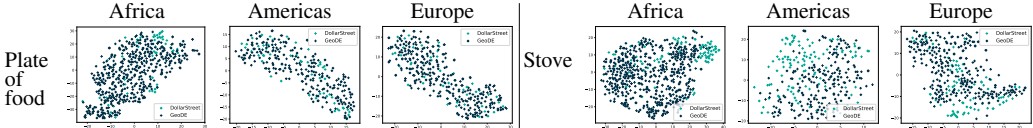

Figure 5: We visualize the TSNE plots for several of object classes per region for DollarStreet (cyan) and GeoDE (dark blue). We see that these features overlap significantly more than that of GeoYFCC, however, there are still objects with different distributions (e.g. "Stove" in Americas).

(Fig. 1) reveals imbalances, with most countries from Europe. Moreover, this dataset does not have curated labels, just tags, and the distribution among tags is also long-tailed (the top 20 of 1,197 tags comprise 34% of the dataset). Comparatively, GeoDE is balanced across both regions and classes. DollarStreet [25] is a much smaller dataset with only 38,479 photos, comparable to GeoDE with 61,940 photos. DollarStreet has more classes but with fewer images per class compared to GeoDE: DollarStreet averages only 133 images per each of its 289 classes (with 382 images on average for its 40 most common classes), while GeoDE average 1,548 images per each of its 40 classes.

**Object appearance.** Finally, we attempt to quantify the differences in the appearance of images collected through different methods, by comparing GeoDE with GeoYFCC [10] and DollarStreet [25]. We extract features for each dataset using a ResNet50 model [17] trained with self-supervised learning SwAV [5] on the PASS dataset [2] (license: CC-BY). We train linear classifiers to predict the dataset given an image; the classifier achieves an accuracy of 96.3% when trying to distinguish between GeoYFCC and GeoDE, and an accuracy of 96.1% when trying to distinguish between DollarStreet and GeoDE. However, this could be the result of having different distributions of regions (in the case of GeoYFCC) and different objects (for both). To understand how the dataset distributions are different beyond just the class/region frequencies we obtain low-dimensional TSNE embeddings [33] with images that restricted to a certain (region, object) pair (Figs. 4 and 5). We see a much more pronounced difference between GeoDE and GeoYFCC, likely due to effects of web-scraping.

## 6 GeoDE as an evaluation dataset

We now analyze the use of GeoDE as an evaluation dataset, by using it to evaluate two canonical models: the recent CLIP [24] and an ImageNet [8]-trained model.

**Implementation details.** For the CLIP model, we use the weights provided for the ViT-B/32 model. We use text prompts for all 40 object categories as described in the zero-shot recognition setup of [24]. To train a model on ImageNet [8], we first match the classes of GeoDE and ImageNet. We find the relevant synsets for each GeoDE class in WordNet [20], and include all images of that synset. For two object categories ("backyard" and "toothpaste/toothpowder") we do not find any matching categories, and so we ignore these categories in the quantitative analysis. We split our filtered ImageNet [8] dataset into train (38,353 images), validation (12,794 images), and test (12,795 images) datasets. As in Sec. 5 we extract features using a ResNet50 model [17] trained with self-supervised learning SwAV [5] on PASS [2], and retrain the final layer.

**Results.** Fig. 8 (*left*) shows the accuracy across different regions on these two models. Both models perform the best on images from Europe and the worst on images from Africa (difference of more than 7% in both cases). Tab. 4 further breaks out the per-object accuracy for CLIP. While the average accuracy is 82.8%, classes like "dustbin" (37.3%), "medicine" (54.1%), "cleaning equipment" (59.0%) and "spices" (63.2%) perform poorly. Fig. 6 shows example errors.

Table 4: Per-class accuracy (in percentage; increasing order) of CLIP [24] on GeoDE. Objects like "dustbin", "medicine" and "cleaning equipment" are poorly recognized, with accuracy as low as 37%.

| dustbin | medicine | clean. equip. | spices | house | tree | waste cont. | candle | toy | backyard | fence | streetlight | stall | lighter | stove | jug | front door | hand soap | plate of food | truck | light fixture | wheelbarrow | storefront | toothpaste | toothbrush | flag | religious bld | bicycle | road sign | hat | cooking pot | boat | hairbrush | car | monument | dog | bag | chair | bus | lightswitch |
|---|---|---|---|---|---|---|---|---|---|---|---|---|---|---|---|---|---|---|---|---|---|---|---|---|---|---|---|---|---|---|---|---|---|---|---|---|---|---|---|
| 37 | 54 | 59 | 63 | 63 | 68 | 69 | 71 | 73 | 74 | 75 | 76 | 76 | 77 | 78 | 85 | 85 | 86 | 88 | 88 | 88 | 89 | 89 | 90 | 90 | 91 | 92 | 92 | 93 | 93 | 95 | 95 | 95 | 96 | 96 | 96 | 96 | 97 | 97 | 98 |

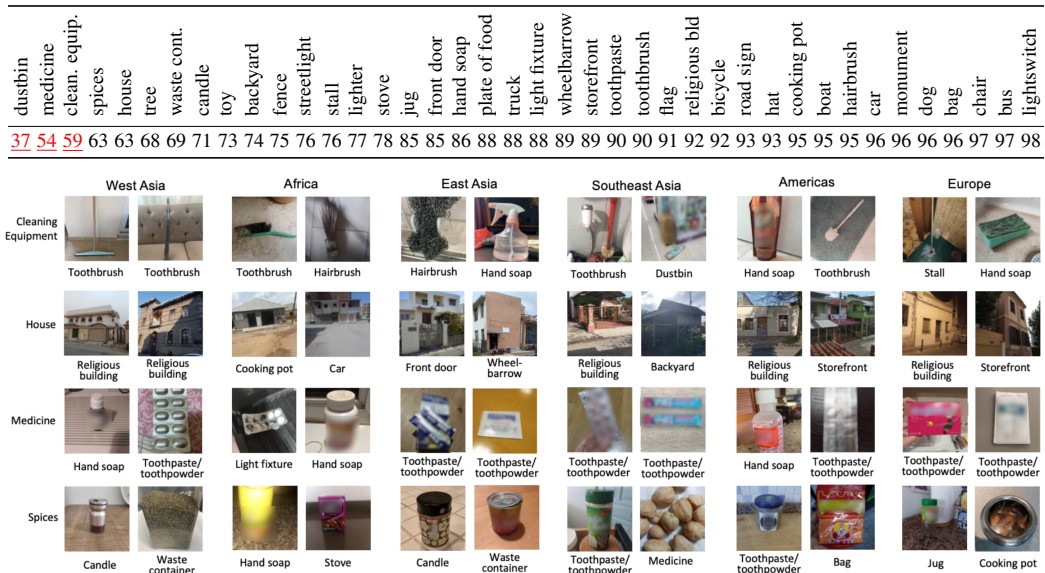

Figure 6: Example errors that the CLIP [24] model makes on GeoDE images (the ground truth label on the left, CLIP prediction at the bottom). There are some systematic errors, e.g., classifying "house" as a "religious building", particularly on images from Asia. (product labels in figure are blurred).

In seeking to understand the accuracy variation across geographic regions, we compute the minimum and maximum per-region accuracy for each object (Fig. 8 *right*). We also compute the confidence interval for the *expected* distribution of per-region accuracy for each object based on the object's overall accuracy.[4] We find that **31** of 40 objects have a minimum and/or maximum region accuracy that falls outside the corresponding 95% confidence interval, suggesting that these objects (including for example "house" and "dustbin") exhibit significant geographic variation (at least with respect to the visual distribution learned by CLIP).[5] We further note some differences on individual classes. "Fence" is 88% accurate for images from Europe, but only 60% and 59% for images from Africa and SE Asia respectively. Similarly, "stove" is 95% accurate in the Americas but only 67% in East Asia. Visualizing classes using TSNE plots of the features (Fig. 7), we see that these objects are region specific, e.g, "religious buildings" from East and SE Asia uniquely include buildings like monasteries and temples; similarly, single- and two-burner "stoves" are primarily from Africa and SE Asia.

## 7 Impact of training with GeoDE

Finally, we attempt to answer how training with GeoDE can improve the performance of models trained on web-scraped data. Concretely, we investigate training a model on jointly GeoDE and subsets of ImageNet [8], and demonstrate the combination improves results across geographic regions.

### 7.1 Training a model with GeoDE

We would like to understand how training a model with data from GeoDE affects object recognition. We train a linear model using pre-trained features on a dataset comprised entirely of ImageNet images and a dataset comprised of both ImageNet and GeoDE images with the same number of images. The feature extractor is a ResNet50 [16] model trained on PASS [2] using SwAV [5]. [6]

**Implementation details.** We split GeoDE into train (4,970 images per region), validation (between 1,657 and 2,188 images per region), and test (between 1,657 and 2,189 images per region). We use

---

[4]We draw 500 random partitions into 6 regions, and compute the resulting per-region accuracies.

[5]Considering the problem of multiple hypothesis testing, we can apply the Bonferonni correction for 40 objects, with $\alpha = 0.05/40 = 0.001$, i.e, 99.9% confidence intervals. Still, **21** of 40 objects fall outside their intervals, confirming that GeoDE as a whole does exhibit statistically significant geographic variation.

[6]We also try a model trained from scratch and with finetuning, with similar conclusions (supp. mat.).

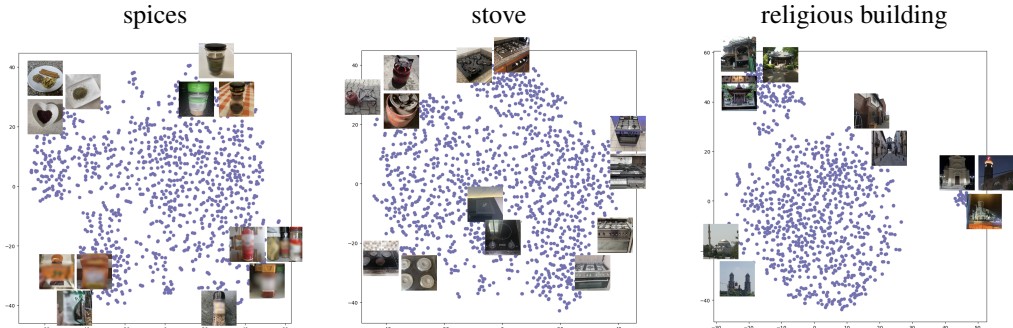

Figure 7: We show the TSNE plots of objects which have large regional disparities in accuracy in the CLIP trained model, with images embedded. We see differences based on region, e.g., "religious buildings" contains a cluster of monasteries and temples, mostly from East and Southeast Asia.

| Model | WAsia | Africa | EAsia | SEAsia | Americas | Europe |
|---|---|---|---|---|---|---|
| ImageNet | 69.4 | 62.7 | 63.3 | 67.3 | 68.6 | **69.9** |
| CLIP | 84.0 | 78.7 | 79.9 | 81.9 | 84.4 | **85.8** |

Figure 8: (*left*) Accuracies (in %) on GeoDE of a model trained on a subset of ImageNet [8] (details in Sec. 6) and of CLIP [24]. The models perform **best** on images from Europe, and worst on images from Africa. (*right*) We compute the maximum and minimum accuracy of CLIP [24] for each object across the 6 regions in GeoDE(sorted by overall accuracy). 31 of 40 objects have at least one region whose accuracy falls outside the 95% CI, suggesting significant differences across regions.

the validation dataset to select training hyperparameters. The training set for our ImageNet only model is the same 38,353 image training set constructed in Sec. 6. To construct the training set of our ImageNet and all regions in GeoDE model, we add in the training sets for all 6 regions in GeoDE while removing the same number of images per class from ImageNet. This procedure gives a training set of 29,820 GeoDE images and 8,533 ImageNet [8] images. The models are trained using an SGD optimizer (lr = 0.1, momentum = 0.9) for 500 epochs with cross entropy loss. Models were trained using a single GPU (RTX1080 or 2080) and took less than 1 hour. Results are reported on the test set.

**Results.** We first report results on the GeoDE test set, and notice a significant improvement in accuracy across all regions, as a result of training with both GeoDE and ImageNet (Tab. 5). However, this improvement could come from the ImageNet + GeoDE dataset matching the domain of the GeoDE evaluation set and may not generalize to other datasets. Thus, we also test these models on a different dataset: the DollarStreet dataset [25]. This dataset has been used before as an evaluation benchmark [7], to understand if current object recognition models can perform well on objects from a diverse set of regions. Tab. 5 lists the accuracy for the 4 different regions in DollarStreet, along with the per class accuracies for the object categories that overlap between GeoDE and DollarStreet. We see an increase in performance across most categories, suggesting that GeoDE is more geo-diverse than ImageNet and that there is an advantage to using geo-diverse data in the training set.

## 7.2 Cost-vs-Diversity tradeoffs

The main drawback of GeoDE is the cost of this dataset: images collected in this way cost more than the standard pipeline of web-scraping and crowd-sourcing annotations. Thus, it is important to identify which classes and regions contribute most to the overall model. To investigate this, we start with the filtered ImageNet dataset described above, vary the amount of GeoDE data from a particular region, and analyze the change in overall regional performance.

**Implementation Details.** We start with the 38,353 filtered ImageNet images and add a region of GeoDE's data back into the dataset and remove the same number of ImageNet images to keep the number of images and class balance the same. Other training details remain the same as in Sec. 7.1.

**Evaluation.** As we are evaluating on the GeoDE test set, there are two possible sources of performance gain: (1) the model is able to take advantage of the additional regional information from the GeoDE data; and (2) the GeoDE images were collected using the same collection method as the test set and from Sec. 5, we saw that there is a difference in the feature space that can be attributed

Table 5: We compare the performance of a model trained on ImageNet [8] versus one that is trained on both ImageNet and GeoDE . We report results on the test set of GeoDE as well as the DollarStreet [25] images. We see an improvement across all regions for both test datasets. We also report the per-class accuracies for the DollarStreet dataset and see improvement across most objects as well.

| | Tested on GeoDE | | | | | | | Tested on DollarStreet [25] | | | | | | | | | | | | | | | | | |
|---|---|---|---|---|---|---|---|---|---|---|---|---|---|---|---|---|---|---|---|---|---|---|---|---|---|
| | West Asia | Africa | East Asia | SE Asia | Americas | Europe | Overall | Africa | America | Asia | Europe | bicycle | chair | clean.equip | cooking pot | dustbin | hand soap | house | light fixture | light switch | medicine | plate of food | stove | toy | Overall |
| ImageNet | 69 | 63 | 63 | 67 | 69 | 70 | 67 | 45 | 64 | 58 | 75 | 92 | 86 | 19 | 49 | **76** | 49 | 88 | 36 | 77 | **80** | 84 | **89** | 50 | 60 |
| +GeoDE | **88** | **87** | **86** | **87** | **89** | **90** | **88** | **55** | **74** | **68** | **80** | **95** | **88** | **36** | **61** | 68 | **65** | **91** | **63** | **79** | 78 | **96** | 85 | **58** | **69** |

| | Classes with largest % inc. in AP |
|---|---|
| *Africa* | waste cont., spices, dustbin, clean. equip. |
| *E. Asia* | relig. blg., spices, dustbin, waste cont. |
| *W. Asia* | dustbin, hand soap, clean. equip., spices, |
| *Americas* | dustbin, spices, clean. equip., medicine |
| *SE. Asia* | waste cont., spices, medicine, clean. equip. |

Figure 9: (*left*) We measure the relative improvement in AP per object when GeoDE images from that region are included in training. Each vertical line represents an object (sorted by region of max. improvement). Africa and East Asia see the largest improvement for the most classes. (*right*) We highlight the classes with the largest increases in the AP when adding in training images from GeoDE.

to the collection method itself (deliberately taking photos rather than web-scraping). In order to distinguish between these two sources, we measure the accuracy on both the region in the train set *and* accuracy on the images from Europe[7]. We also measure the increase in AP for specific objects to better understand which objects benefit most from GeoDE data.

**Results.** We find that the performance within each specific region and in Europe increase with the additional GeoDE data. The relative increase in performance for the specific region is larger than the increase for Europe, showing the value of data for each region, moreover, the improvements do not saturate, suggesting that more data could lead to further gains. Full results are presented in the supp. mat. We examine the classes that have the largest increase in average precision (AP) as the regional GeoDE images are added to the dataset in Fig. 9 (*left*). We also present the object classes that see the most improvement in Fig. 9 (*right*). In general, we see that specific objects such as "spices", "waste container" and "cleaning equipment" benefit most from regional GeoDE data.

## 8   Conclusion

We introduced a new dataset GeoDE which uses crowd-sourcing for image collection, a significant departure from the popular computer vision dataset collection paradigm of web-scraping for image collection. Through this collection method, we ensured that this dataset does not contain personally identifiable information, we own the rights to the images, the image creators were compensated for their work, and were able to control for geographic diversity and object distribution in the dataset. We show that GeoDE is a useful dataset for highlighting shortcomings in common models (e.g., CLIP) and can improve performance when added to the training dataset. Also, GeoDE shows that crowd-sourcing is a viable image collection method for creating diverse and responsible datasets.

**Acknowledgements.** This material is based upon work partially supported by the National Science Foundation under Grant No. 2145198. Any opinions, findings, and conclusions or recommendations expressed in this material are those of the author(s) and do not necessarily reflect the views of the National Science Foundation. We also acknowledge support from Meta AI and the Princeton SEAS Howard B. Wentz, Jr. Junior Faculty Award to OR. We thank Dhruv Mahajan for his valuable insights during the project development phase. We also thank Jihoon Chung, Nicole Meister, Angelina Wang and the Princeton Visual AI Lab for their helpful comments and feedback during the writing process.

---

[7]We use Europe as this region had the best performance when using a model trained on just ImageNet.

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
