# GeoDE: a Geographically Diverse Evaluation Dataset for Object Recognition

**Vikram V. Ramaswamy**[1], **Sing Yu Lin**[1], **Dora Zhao**[2]*, **Aaron B. Adcock**[3],
**Laurens van der Maaten**[3], **Deepti Ghadiyaram**[4], **Olga Russakovsky**[1]

[1]Princeton University    [2]Sony AI    [3]Meta AI    [4]Runway
*Work done as a graduate student at Princeton University

1 Here, we provide some more details about our experiments.

2 • In Sec A, we provide the datasheet for GeoDE , as in [4].

3 • In Sec B, we describe our heuristic to select object categories in more detail.

4 • In Sec C, we compare the GeoDE feature space to that of ImageNet [2]

5 • In Sec. E, we provide results when finetuning pre-trained models rather than just training
6 the final layer of a ResNet.

7 • In Sec. F, we give more details about GeoDE , including the counts of images of different
8 regions and categories, as well as more sample images from this dataset.

## A Datasheet for GeoDE

10 We include the datasheet for GeoDE below, based on Datasheets for Datasets [4][1]

---

### MOTIVATION

11 **For what purpose was the dataset created?** Was there a specific task in mind? Was there a specific
12 gap that needed to be filled? Please provide a description.
13 GeoDE was created for 2 purposes: (1) To construct a more geographically diverse dataset for training
14 and evaluation and (2) to understand what it would take to crowd-source an image dataset from scratch.
15

16 **Who created this dataset (e.g., which team, research group) and on behalf of which entity (e.g.,**
17 **company, institution, organization)?**
18 GeoDE was created via a collaboration between the Princeton Visual AI lab and Meta research.
19

20 **What support was needed to make this dataset?** (e.g.who funded the creation of the dataset? If
21 there is an associated grant, provide the name of the grantor and the grant name and number, or if it
22 was supported by a company or government agency, give those details.)
23 Creation of GeoDE was partially supported by the National Science Foundation under Grant No.
24 2145198. It was also supported by Meta AI and the Princeton SEAS Howard B. Wentz, Jr. Junior
25 Faculty Award to OR.

26 **Any other comments?**
27 N/A
28

---

[1]The template used is from `https://github.com/AudreyBeard/Datasheets-for-Datasets-Template`

Submitted to the 37th Conference on Neural Information Processing Systems (NeurIPS 2023) Track on Datasets and Benchmarks. Do not distribute.

## COMPOSITION

**What do the instances that comprise the dataset represent (e.g., documents, photos, people, countries)?** Are there multiple types of instances (e.g., movies, users, and ratings; people and interactions between them; nodes and edges)? Please provide a description.

GeoDE consists of images of 40 different categories from 6 different regions. These categories and regions are fully listed in tables 2 and 3 of the main paper. Additionally, meta data for each image includes the type of phone used to take the picture, the GPS coordinates of the image, whether there are people present in the background of the image (note that there are no recognizable people in the dataset), and if a large fraction of the image consists of trees.

**How many instances are there in total (of each type, if appropriate)?**
There are 61, 940 images.

**Does the dataset contain all possible instances or is it a sample (not necessarily random) of instances from a larger set?** If the dataset is a sample, then what is the larger set? Is the sample representative of the larger set (e.g., geographic coverage)? If so, please describe how this representativeness was validated/verified. If it is not representative of the larger set, please describe why not (e.g., to cover a more diverse range of instances, because instances were withheld or unavailable).

GeoDE consists of a sample of images. The larger set would include all possible objects from all possible regions of the world. GeoDE is balanced across 6 regions and 40 objects,(regions chosen to maximise geodiversity).

**What data does each instance consist of?** "Raw" data (e.g., unprocessed text or images) or features? In either case, please provide a description.

Each instance consists of an image, location, object in the image, whether there are people in the background, whether a significant portion of the image contains trees.

**Is there a label or target associated with each instance?** If so, please provide a description.
Yes, each image is labelled with one of 40 objects.

**Is any information missing from individual instances?** If so, please provide a description, explaining why this information is missing (e.g., because it was unavailable). This does not include intentionally removed information, but might include, e.g., redacted text.
No

**Are relationships between individual instances made explicit (e.g., users' movie ratings, social network links)?** If so, please describe how these relationships are made explicit.
N/A

**Are there recommended data splits (e.g., training, development/validation, testing)?** If so, please provide a description of these splits, explaining the rationale behind them.
No, we created training, validation and test splits by randomly splitting the dataset.

**Are there any errors, sources of noise, or redundancies in the dataset?** If so, please provide a description.
Possible errors include issues with labels of objects and countries. To the best of our knowledge this is limited to lesser than 1% of the dataset.

**Is the dataset self-contained, or does it link to or otherwise rely on external resources (e.g., websites, tweets, other datasets)?** If it links to or relies on external resources, a) are there guarantees that they will exist, and remain constant, over time; b) are there official archival versions of the complete dataset (i.e., including the external resources as they existed at the time the dataset was created); c) are there any restrictions (e.g., licenses, fees) associated with any of the external resources that might apply to a future user? Please provide descriptions of all external resources and any restrictions associated with them, as well as links or other access points, as appropriate.

GeoDE is self contained

**Does the dataset contain data that might be considered confidential (e.g., data that is protected by legal privilege or by doctor-patient confidentiality, data that includes the content of individuals' non-public communications)?** If so, please provide a description.

No, GeoDE does not contain confidential data.

**Does the dataset contain data that, if viewed directly, might be offensive, insulting, threatening, or might otherwise cause anxiety?** If so, please describe why.

No

**Does the dataset relate to people?** If not, you may skip the remaining questions in this section.

No

**Does the dataset identify any subpopulations (e.g., by age, gender)?** If so, please describe how these subpopulations are identified and provide a description of their respective distributions within the dataset.

N/A

**Is it possible to identify individuals (i.e., one or more natural persons), either directly or indirectly (i.e., in combination with other data) from the dataset?** If so, please describe how.

N/A

**Does the dataset contain data that might be considered sensitive in any way (e.g., data that reveals racial or ethnic origins, sexual orientations, religious beliefs, political opinions or union memberships, or locations; financial or health data; biometric or genetic data; forms of government identification, such as social security numbers; criminal history)?** If so, please provide a description.

N/A

**Any other comments?**

N/A

---

## COLLECTION

**How was the data associated with each instance acquired?** Was the data directly observable (e.g., raw text, movie ratings), reported by subjects (e.g., survey responses), or indirectly inferred/derived from other data (e.g., part-of-speech tags, model-based guesses for age or language)? If data was reported by subjects or indirectly inferred/derived from other data, was the data validated/verified? If so, please describe how.

Participants from across the world took photos of different objects and submitted it. They were compensated for their efforts. This data was manually checked and verified to contain the object.

**Over what timeframe was the data collected?** Does this timeframe match the creation timeframe of the data associated with the instances (e.g., recent crawl of old news articles)? If not, please describe the timeframe in which the data associated with the instances was created. Finally, list when the dataset was first published.

Data was collected in 2022. It was first publicly released in January 2023.

**What mechanisms or procedures were used to collect the data (e.g., hardware apparatus or sensor, manual human curation, software program, software API)?** How were these mechanisms or procedures validated?

We used manual human curation: participants took photos of different objects. All images were verified by Appen's quality analysis team.

**What was the resource cost of collecting the data?** (e.g. what were the required computational resources, and the associated financial costs, and energy consumption - estimate the carbon footprint. See Strubell *et al.*[9] for approaches in this area.)

Total cost for all images was $54,000, not including researcher time. There were no models involved in collecting the dataset.

**If the dataset is a sample from a larger set, what was the sampling strategy (e.g., deterministic, probabilistic with specific sampling probabilities)?**

It is not a sample from a larger dataset.

**Who was involved in the data collection process (e.g., students, crowdworkers, contractors) and how were they compensated (e.g., how much were crowdworkers paid)?**

Participants from across the world were tasked with taking photos of specific objects and paid for their time. We partnered with Appen (`www.appen.com`), who recruited and compensated the workers. Compensation varied depending on the region, but we got assurances that the pay was appropriate for the work.

**Were any ethical review processes conducted (e.g., by an institutional review board)?** If so, please provide a description of these review processes, including the outcomes, as well as a link or other access point to any supporting documentation.

No,

**Does the dataset relate to people?** If not, you may skip the remainder of the questions in this section.

No

**Did you collect the data from the individuals in question directly, or obtain it via third parties or other sources (e.g., websites)?**

N/A

**Were the individuals in question notified about the data collection?** If so, please describe (or show with screenshots or other information) how notice was provided, and provide a link or other access point to, or otherwise reproduce, the exact language of the notification itself.

N/A

**Did the individuals in question consent to the collection and use of their data?** If so, please describe (or show with screenshots or other information) how consent was requested and provided, and provide a link or other access point to, or otherwise reproduce, the exact language to which the

individuals consented.

N/A

**If consent was obtained, were the consenting individuals provided with a mechanism to revoke their consent in the future or for certain uses?** If so, please provide a description, as well as a link or other access point to the mechanism (if appropriate)

N/A

**Has an analysis of the potential impact of the dataset and its use on data subjects (e.g., a data protection impact analysis)been conducted?** If so, please provide a description of this analysis, including the outcomes, as well as a link or other access point to any supporting documentation.

N/A

**Any other comments?**

N/A

## PREPROCESSING / CLEANING / LABELING

**Was any preprocessing/cleaning/labeling of the data done(e.g.,discretization or bucketing, tokenization, part-of-speech tagging, SIFT feature extraction, removal of instances, processing of missing values)?** If so, please provide a description. If not, you may skip the remainder of the questions in this section.

No.

**Was the "raw" data saved in addition to the preprocessed/cleaned/labeled data (e.g., to support unanticipated future uses)?** If so, please provide a link or other access point to the "raw" data.

N/A

**Is the software used to preprocess/clean/label the instances available?** If so, please provide a link or other access point.

N/A

**Any other comments?**

N/A

## USES

**Has the dataset been used for any tasks already?** If so, please provide a description.

GeoDE has been used to evaluate large scale models for geographical bias in this paper.

**Is there a repository that links to any or all papers or systems that use the dataset?** If so, please provide a link or other access point.

No.

**What (other) tasks could the dataset be used for?**

Additional uses of GeoDE could be trying to understand geodiversity of current datasets, to understand how webscraping and crowd-collected images differ (a small analysis done in section 5 of the main paper).

**Is there anything about the composition of the dataset or the way it was collected and preprocessed/cleaned/labeled that might impact future uses?** For example, is there anything that a future user might need to know to avoid uses that could result in unfair treatment of individuals or groups (e.g., stereotyping, quality of service issues) or other undesirable harms (e.g., financial harms, legal risks) If so, please provide a description. Is there anything a future user could do to mitigate these undesirable harms?

All images in GeoDE were collected by participants with smart phones. Thus, the dataset does not exhibit economic diversity.

**Are there tasks for which the dataset should not be used?** If so, please provide a description.

N/A

**Any other comments?**

N/A

---

## DISTRIBUTION

**Will the dataset be distributed to third parties outside of the entity (e.g., company, institution, organization) on behalf of which the dataset was created?** If so, please provide a description.

Yes, GeoDE is freely available to download at `https://geodiverse-data-collection.cs.princeton.edu/`

**How will the dataset will be distributed (e.g., tarball on website, API, GitHub)?** Does the dataset have a digital object identifier (DOI)?

GeoDE is available as a .zip file to download.

**When will the dataset be distributed?**

GeoDE is currently available.

**Will the dataset be distributed under a copyright or other intellectual property (IP) license, and/or under applicable terms of use (ToU)?** If so, please describe this license and/or ToU, and provide a link or other access point to, or otherwise reproduce, any relevant licensing terms or ToU, as well as any fees associated with these restrictions.

No, GeoDE is released under a CC-BY license.

**Have any third parties imposed IP-based or other restrictions on the data associated with the instances?** If so, please describe these restrictions, and provide a link or other access point to, or otherwise reproduce, any relevant licensing terms, as well as any fees associated with these restrictions.

No

**Do any export controls or other regulatory restrictions apply to the dataset or to individual instances?** If so, please describe these restrictions, and provide a link or other access point to, or otherwise reproduce, any supporting documentation.

No

**Any other comments?**

N/A

**Who is supporting/hosting/maintaining the dataset?**
Currently, GeoDE is being hosted by the Princeton computer science department, specifically, Dr. Vikram Ramaswamy and Prof. Olga Russakovsky. For the long term, we are considering one of two options: partnering with Common Visual Data Foundation (CVDF; `http://www.cvdfoundation.org/`) or utilizing `https://researchdata.princeton.edu/news/2023-05-25/coming-soon-princeton-data-commons` (we've seen internal versions which look great for our use cases but are waiting for it to become public)

**How can the owner/curator/manager of the dataset be contacted (e.g., email address)?**
Questions can be emailed to vr23@cs.princeton.edu

**Is there an erratum?** If so, please provide a link or other access point.
No.

**Will the dataset be updated (e.g., to correct labeling errors, add new instances, delete instances)?** If so, please describe how often, by whom, and how updates will be communicated to users (e.g., mailing list, GitHub)?
Yes, the dataset will be updated as needed, by Vikram Ramaswamy. Updates will be posted on the GitHub repo along with the website, on how to access the corrected version.

**If the dataset relates to people, are there applicable limits on the retention of the data associated with the instances (e.g., were individuals in question told that their data would be retained for a fixed period of time and then deleted)?** If so, please describe these limits and explain how they will be enforced.
N/A

**Will older versions of the dataset continue to be supported/hosted/maintained?** If so, please describe how. If not, please describe how its obsolescence will be communicated to users.
No, older versions will not continue to be hosted, however, we will provide information on our github as well as the webpage, with a script to update the dataset (if applicable).

**If others want to extend/augment/build on/contribute to the dataset, is there a mechanism for them to do so?** If so, please provide a description. Will these contributions be validated/verified? If so, please describe how. If not, why not? Is there a process for communicating/distributing these contributions to other users? If so, please provide a description.
No, there is no current mechanism to do so. Users can provide feedback / corrections to the dataset on github, which we will use to update the dataset.

**Any other comments?**
N/A

## B Selecting object categories for GeoDE

In this section, we provide more details about the object selection heuristic we employed. We mainly used the GeoYFCC [3] dataset that was constructed to be geodiverse.

| **Leave one out training** |
| --- |
| curler, fan, footstool, chili, coconut, toilet, canoe, motorboat, mountain-bike, stupa, villa, backpack, baseball-glove, basin, basket, bat, bathtub, battery, beer-mug, belt, blade, bowl, bowl, broom, bucket, carryall, case, cash-machine, cassette, cleaver, cologne, cooler, counter, dinner-dress, dinner-jacket, dish, gown, grille, hammer, jacket, kettle, microphone, parka, porch, rack, remote-control, sandal, scale, shelf, shot-glass, stereo, stocking, stool, sweater, tape, teddy, timer, tripod, trouser, turntable, wardrobe, weight, wok, woodcarving, hot-pot, chewing-gum, cucumber, lime, fig, pineapple, jackfruit, kiwi, mango, basil, garlic, sage, lager, ale, porter, stout, champagne, rum, tequila, vodka, whiskey, mocha |
| **Linear SVM for region** |
| mountain-bike, bicycle, raft, ferry, ship, kayak, streetcar, bus, impala, car, footstool, bench, chair, mushroom, breakfast, vegetable, dessert, dinner, door, bowler-hat, house, building, chandelier, lamp, light, castle, acropolis, fortress, tower, palace, dome, architecture, memorial, statue, sculpture, gravestone, arch, temple, stupa, monastery, church, cathedral, chapel, mosque, signboard, grocery-store, shop, kitchen, lantern, doll, coati, cork, primate, alp, shore, curler, cologne, seashore, gnu, hog, giraffe, arctic, ice-rink, ski, elephant, guinness, makeup, circuit, geyser, skyscraper, hippopotamus, basketball, paintball, sword, hijab, fortification, craft, clock, stage, tractor, dagger, defile, bikini, swing, windmill, motor, brick, snowboard, course, volleyball, display, opera, railing, playground, veranda, wind-instrument, city-hall, ruin, portfolio, newspaper, airbus, bridge, airfield, global-positioning-system, brake-drum, kid, mangrove, motor-scooter, crane, intersection, plain, column, wardrobe, interface, guitar, costume, grand-piano, aircraft, factory, seaside, ball, sweet, gravy-boat, spotlight, american-bison, sail, beer, pier, road, tulip, grass, miniskirt, willow, flood, street, roof, slide, cliff, track, train, vehicle, boot, world, patio, window, rainbow, beacon, sidewalk, organ-pipe, tank, cable-car, grey, hall, map, cattle, airport, school, mountain, promontory, monkey, motorcycle, bubble, black, mirror, golf-club, skateboard, computer, university, denim, sky, rock, earphone, descent, garden, hill, library, tea, blush-wine, radio, bill, sunglass, ballpark, apparel, web, field-glass, reef, fountain, downhill, pen, cable, step, graffito, conveyance, fabric, hovel, umbrella, iron, cloud, strand, toilet, walker, valley, airplane, cup, base, wire, camel, pizza, bathroom, lounge, dock, van, circuit-board, bell, sheep, book, fish, canyon, fire, array, rangefinder, coca-cola |
| **Clustering** |
| acropolis, cork, coati, footstool, stupa, impala, chili, primate, cologne, gnu, guinness, alp, hog, shore, boater, walker, plain, hippopotamus, raft, chandelier, curler, giraffe, arctic, bowler-hat, castle, geyser, boot, streetcar, rum, hijab, ski, temple, windmill, dagger, fortification, snowboard, coffee, ice-rink, display, cathedral, bench, bikini, lantern, slope, elephant, strand, sword, paintball, gravestone, tulip, golf-club, downhill, swing, volleyball, mushroom, monastery, american-bison, stage, cup, church, wardrobe, wind-instrument, skyscraper, sweet, course, tower, opera, sketch, circuit, chapel, col, motor, clock, railing, mangrove |

Table 1: Prospective tags identified from GeoYFCC [3]. Tags in red seemed hard to picture. Tags in blue are of animals that might be hard to crowdsource

.

**Implementation details.** Features for GeoYFCC were extracted using a ResNet50 [5] pretrained on ImageNet [2]. We used Logistic regression, Linear SVM and KMeans clustering implementations from the sklearn library [6]. We used continents as regions. GeoYFCC [3] contains over 1200 tags, we ignored all tags with counts in the bottom 20th percentile, giving us a total of 745 tags.

First, we apply each of these methods to GeoYFCC to identify candidate tags.

- For each region $R$, we train a linear model using a feature extractor and images from all regions except $R$ and a linear model trained on all images from all regions, to predict the presence or absence of each tag. We then applied both models to images from $R$. The difference in performance between these models allows us to measure the difference in appearance of the tag. We selected tags where in the weighted average precision on the region was less than 0.8* the performance on other regions. This gave us a set of 277 tags such as "footstool", "chili", "case", etc.
- For each tag $T$, we train a linear SVM to predict the region given the features of images containing tag $T$. If this model has high accuracy, this suggests that this tag is visually different across regions. We selected tags that had an accuracy of over 50%. 223 tags were identified in this manner. "Cork", "bowler_hat" and "mountain_bike" are examples of tags found in this way.
- We clustered features of images containing tag $T$. We then computed the Gini impurity of each world region, and selected tags that had a median Gini value of at least 0.5. This gave us 75 tags in total. Examples of tags found in this way were "chili", "footstool" and "stove".

After identifying these tags, we first pruned them by removing tags that did not appear to correspond to an object. Examples of this include "arctic", "descent", etc. Second, we removed tags corresponding to wild animals, since these would not be found in all regions. Examples of tags removed in this

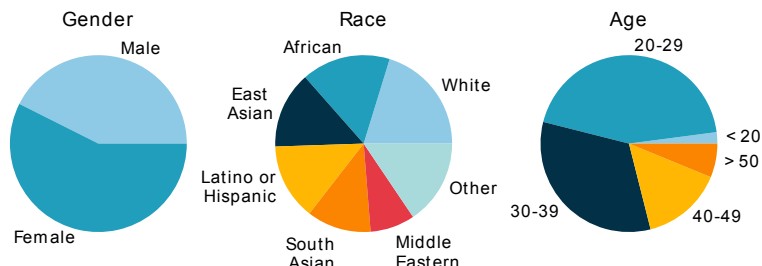

Figure 1: Participant demographics

manner were "gnu", "camel", etc. This gave us a list of 265 tags. Third, these tags were sometimes variants of objects, for example, we had tags like "stupa", "temple", "church", "mosque" and "chapel". Thus we grouped tags based on meaning. Other examples included "breakfast", "dinner" and "dessert" as "plate of food", "stool", "footstool" and "bench" as "chair", etc. This gave us a list of objects we could collect, e.g. "religious buildings", "plate of food", "toy".

We also provide the user demographics of the participants (Fig. 1). As shown, images in GeoDE were provided by people of varying genders, ethnicities and ages.

## C  Comparison with ImageNet

In this section, we run more comparisons of GeoDE with ImageNet [2] and DollarStreet [8].

### C.1  Comparison to ImageNet

We note that the comparison to GeoYFCC [3] in Sec. 5 in the main text required us to use tags which are noisy. Thus, to compare GeoDE to another web-scraped dataset, we compare GeoDE to ImageNet in this section, checking how much the feature spaces differ.

We find a subset of ImageNet21k as outlined in Sec. 6, and extract features using a PASS [1] trained ResNet50 [5] model. Other implementation details remain the same as in Sec 5 in the main paper.

We first use a Logistic regression model to predict the dataset that the features are taken from and this has an accuracy of 96.0%, showing that the feature space is very different. We also visualize TSNE plots of different objects in figure 2.

### C.2  Comparison with DollarStreet

Compared to GeoDE, DollarStreet [8] contains a lot more categories, thus resulting in much fewer images per category (on average, the top 40 object categories in DollarStreet contain only 382 images, whereas GeoDE has an average of 1548 images per category). Thus, when filtering images to comprise of only common categories between both datasets, we end up with much fewer images for DollarStreet. We run small scale tests to understand how these images differ from those in GeoDE.

**Relative value of an image.** Similar to the canonical work on dataset bias [10], we measure the relative value of an image from GeoDE and DollarStreet. That is, we measure the number of training images needed for strong cross-dataset generalization. Concretely, we select 13 classes from DollarStreet which (1) appear in GeoDE, and (2) have more than $100$ images. We restrict both datasets to these 13 classes, resulting in $4, 788$ images for DollarStreet and $17, 245$ images for GeoDE. We now extract features using a PASS trained network, and train linear models to predict the 13 classes. First, we train a baseline model on 250 randomly sampled DollarStreet images and evaluate it on the remaining DollarStreet images; we then train models with increasing numbers of images from GeoDE until we match the accuracy of the DollarStreet-trained baseline. This occurs with 3,000 GeoDE training images. Next, we do a similar process for a baseline trained on 250

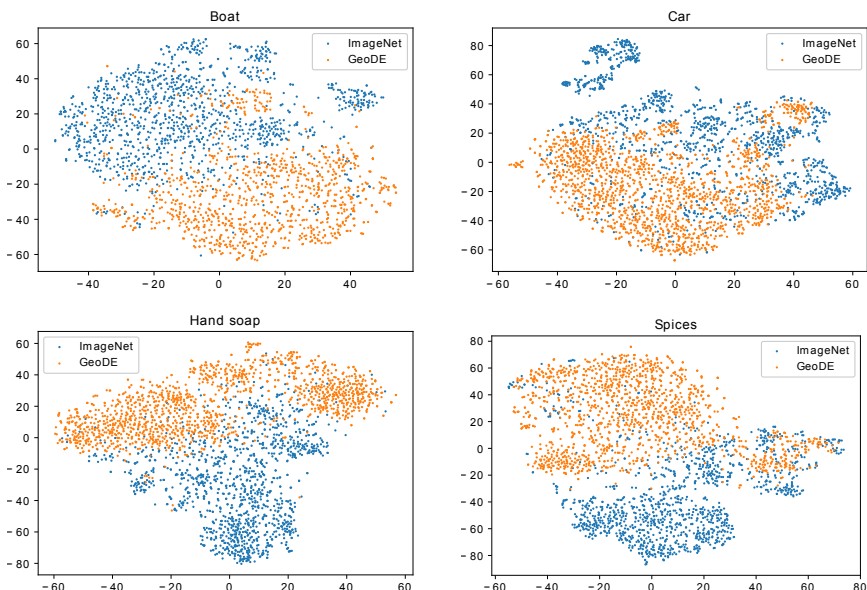

Figure 2: We visualize the TSNE plots for several of object classes using ImageNet and GeoDE . While the features do overlap slightly, on the whole, they are very different for dataset distributions, even within each category.

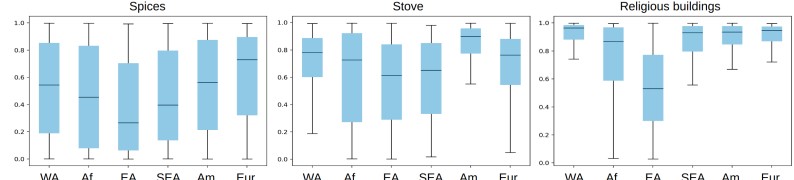

Figure 3: We visualize the probabilities assigned by CLIP to different images from the same class as a box plot. We see that the images across different regions have scores that vary in different ways: for "spices", we see a large variance for all regions; for "stoves", the variance is large for stoves from Africa, and Southeast and East Asia, but much smaller for other regions. For "religious buildings", we see that the scores are just much lower for buildings in East Asia.

randomly sampled GeoDE images. However, we are unable to match its accuracy on GeoDE using DollarStreet training images, even after using all $4,788$ images of these classes, showing a higher relative value per image in GeoDE .

## D    More results using GeoDE as an evaluation dataset

Here, we provide more more analysis performed on using GeoDE as an evaluation dataset. As shown in sec. 5 of the main text, CLIP models perform worse on certain objects (e.g "house", "spices", "medicine", etc.). Visualizing the probabilities assigned to different images from the same class as a box plot, we see different scenarios emerge: the variance of scores is large for all regions, as in the case of spices; the variance is large for certain regions, as in the case of stoves from Africa and East and Southeast Asia, or the scores are much lower for a specific region, as in the case of religious buildings in East Asia (3).

## E    More results when training with GeoDE

In this section, we provide results for the incremental training with GeoDE for different regions that were not presented in Sec 7, and provide results when fine-tuning a ResNet50 model, rather than freezing the layer weights.

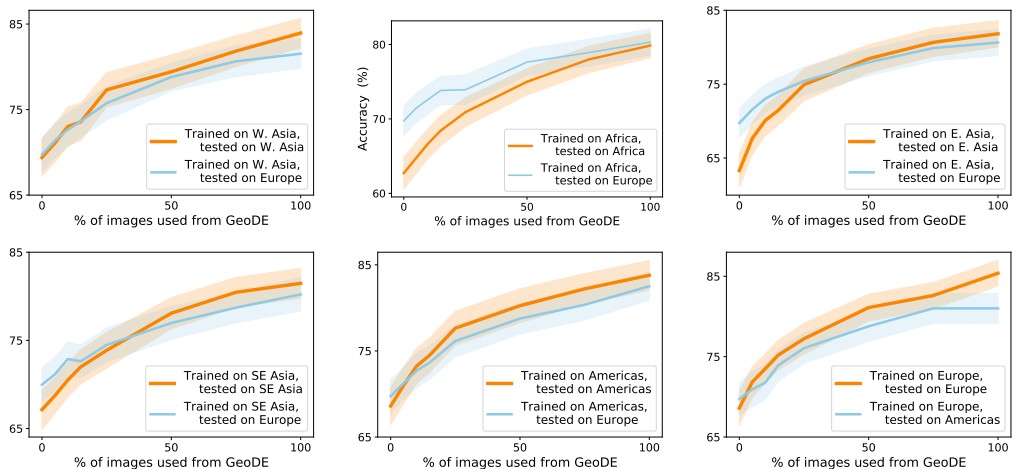

Figure 4: We visualize the increase in accuracy as images are incrementally added in from a region. We find that while adding any GeoDE regional images increases the performance of the model on European images, it has a larger effect on the region the images were drawn from.

## E.1 Results from incrementally adding additional regions

We visualize the improvement in the accuracy as we incrementally add in images from different regions (Sec. 6.2 in the main text). We can see that the performance both within the specific region and in Europe (compared to Americas when considering Europe) increase with the additional GeoDE data. We see that the increase within the region is larger than that of the control, showing that these images are from different domains. (Fig. 4)

## E.2 Results from finetuning a ResNet50 model

**Implementation details.** We use a ResNet50 [5] model pretrained on Imagenet and fine tune the weights using different fractions of the ImageNet and GeoDE datasets as mentioned in Sec. 7 in the main paper. We train the model with an SGD optimizer, learning rate = 0.1, and momentum=0.9. Other implementation details remain the same as before.

**Results.** While the overall trend of the results are the same, we see that these results are slightly noisier, potentially because the model overfits to the small training set (Fig. 5).

## E.3 Performance on individual countries

We measure the performance of models on individual countries within each region of GeoDE. We use two different models: first, we use the CLIP [7] model to understand performance of current models on GeoDE. Next, we train a simple linear model of features extracted from PASS [1] for the GeoDE dataset, and measure performance on individual countries. Implementation details for the CLIP model are the same as in Sec. 6 in the main paper, implementation details for the GeoDE model are as in section 7.1 of the main paper. Accuracies are computed for all countries with at least 25 images in the test set.

Tab. 2 summarizes our results for both. When using a pre-trained model like CLIP, we do notice differences between accuracies of countries within each region, however, we note that the overall trend in accuracy remains roughly the same. When training with images from GeoDE, discrepancies between countries within each region further reduces.

## F    More details about the dataset

In this supplementary section, we provide counts of the objects per region in GeoDE as well as more examples of images from this dataset.

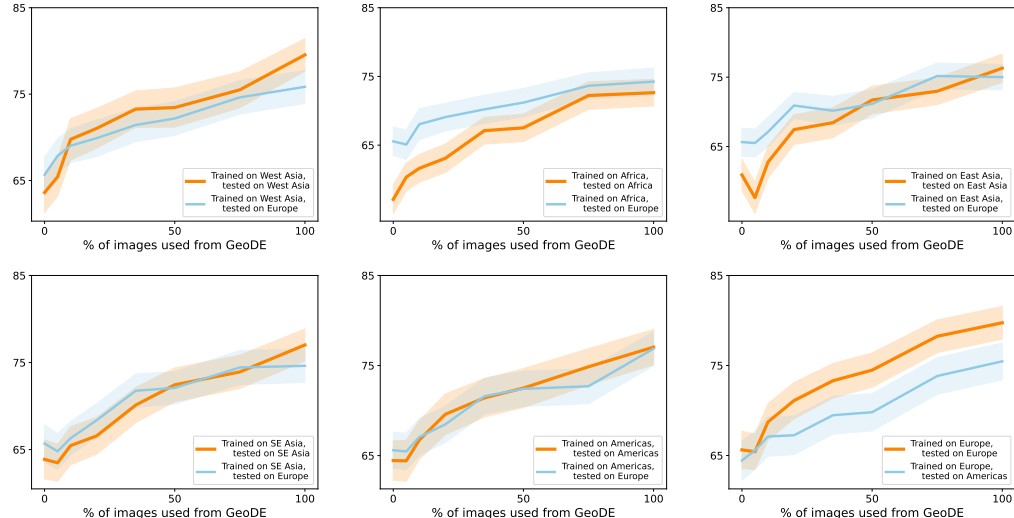

Figure 5: We visualize the increase in accuracy as images are incrementally added in from a region when finetuning a ResNet50 model. Similar to Fig. 10 in the main text, we see that adding in GeoDE images increases performance more in the region than a control.

| | CLIP [7] model | | | | Train on GeoDE | | | |
|---|---|---|---|---|---|---|---|---|
| | Average | STD | Minimum | Country | Average | STD | Minimum | Country |
| Africa | 79.4 | 2.9 | 75.4 | Nigeria | 86.7 | 1.6 | 84.5 | Egypt |
| Americas | 84.4 | 0.2 | 84.1 | Argentina | 88.8 | 1.0 | 87.7 | Argentina |
| EastAsia | 80.2 | 2.0 | 77.2 | China | 89.0 | 1.1 | 87.9 | Japan |
| Europe | 85.9 | 3.6 | 78.3 | Portugal | 91.9 | 1.8 | 89.2 | United Kingdom |
| SouthEastAsia | 82.7 | 1.1 | 81.4 | Indonesia | 88.3 | 1.0 | 87.0 | Indonesia |
| WestAsia | 82.3 | 2.0 | 79.1 | Jordan | 88.9 | 3.6 | 84.2 | Saudi Arabia |

Table 2: We measure the performance of a CLIP based model as well as model trained on GeoDE on individual countries within each region. For each region, we report the average accuracy, the standard deviation, and the country with the minimum accuracy for all countries with over 25 images. We find that while a pretrained model does show significant discrepancies among countries within the same region, training on GeoDE data does reduce this.

As mentioned before, GeoDE is mostly balanced across both region and object: for most part, we were able to get atleast 150 images per region per object, with a few exceptions ("wheelbarrow" in 2 regions; "monument", "boat" and "flag" in 1 region). See Tab. 3 for full counts.

We also provide more examples of the images from GeoDE in the Figures Figs. 6 to 19.

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

**Backyard**

West Asia

Africa

East Asia

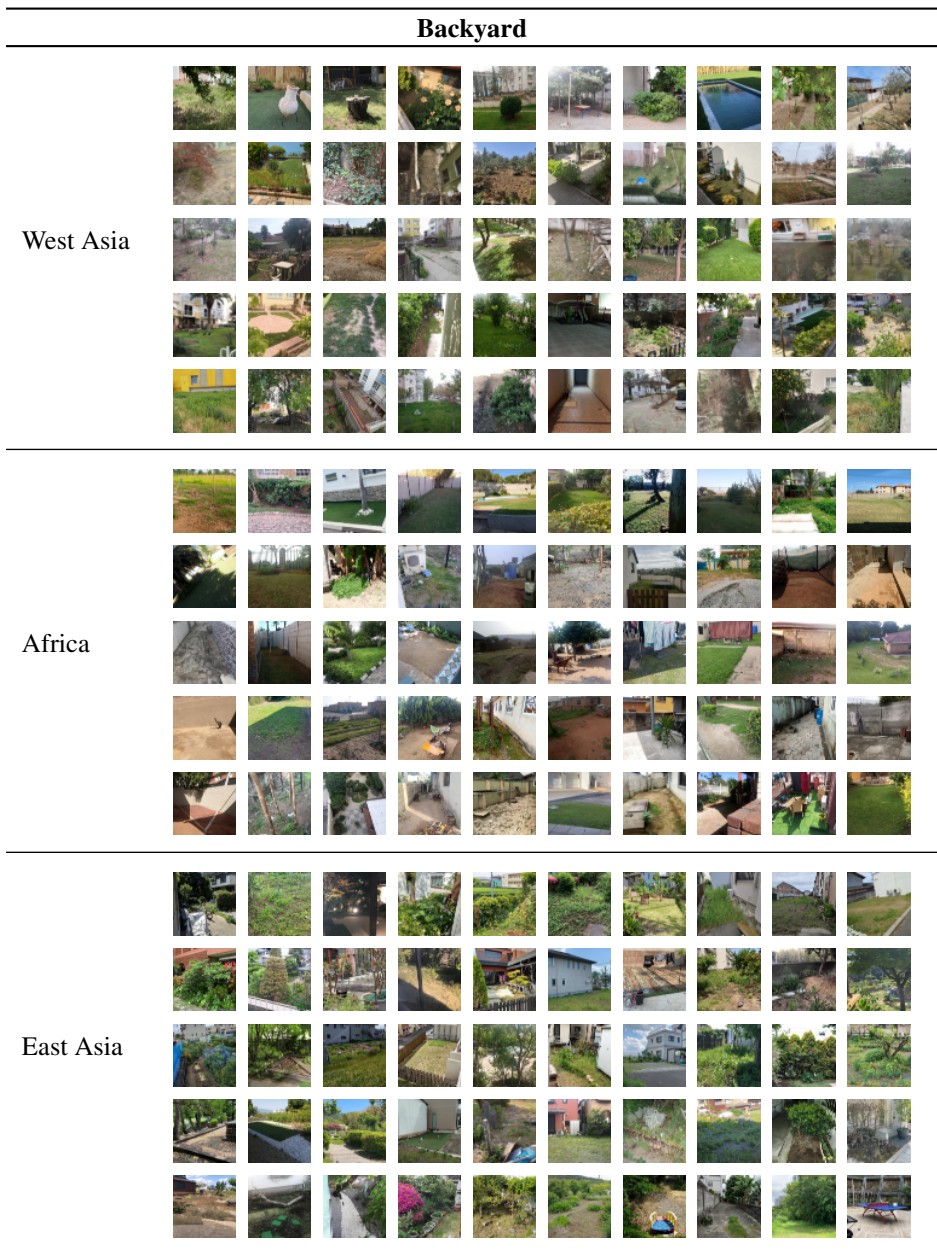

Figure 6: Randomly chosen images for "backyard" for 3 regions. We notice that some of these are backyards made of concrete (West Asia: r2c1, r3c4, etc., Africa: r3c1 ,r4c9, r5c3, etc.,) or do not contain lawns (West Asia: r3c1, r2c5, etc., Africa:r5c1-5, etc., East Asia: r2c4, r5c1, r5c4-6, etc.)

**Backyard**

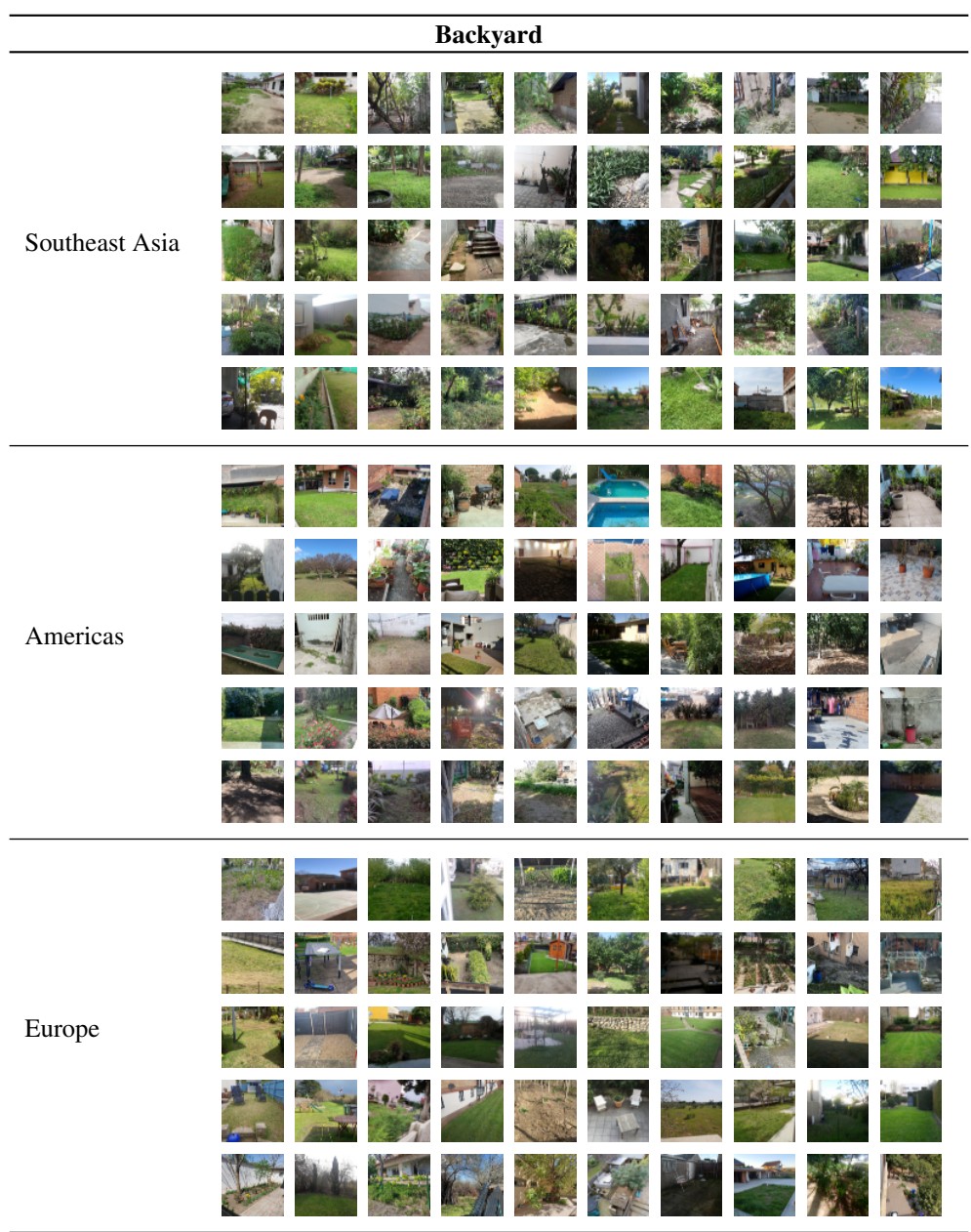

Figure 7: Randomly chosen images for "backyard" for the 3 other regions. Again, we see that regions tend to have backyards made of concrete or paved (Southeast Asia: r1c8, r2c5, r3c3 as examples, Americas: r1c9, r1c10, r2c10, etc., Europe: r3c2, r4c5, etc ), or do not contain lawns (Southeast Asia: r1c1, r1c9, etc., Americas:r3c2, r3c3, etc., Europe: r3c2, r5c9, etc.)

**Bicycle**

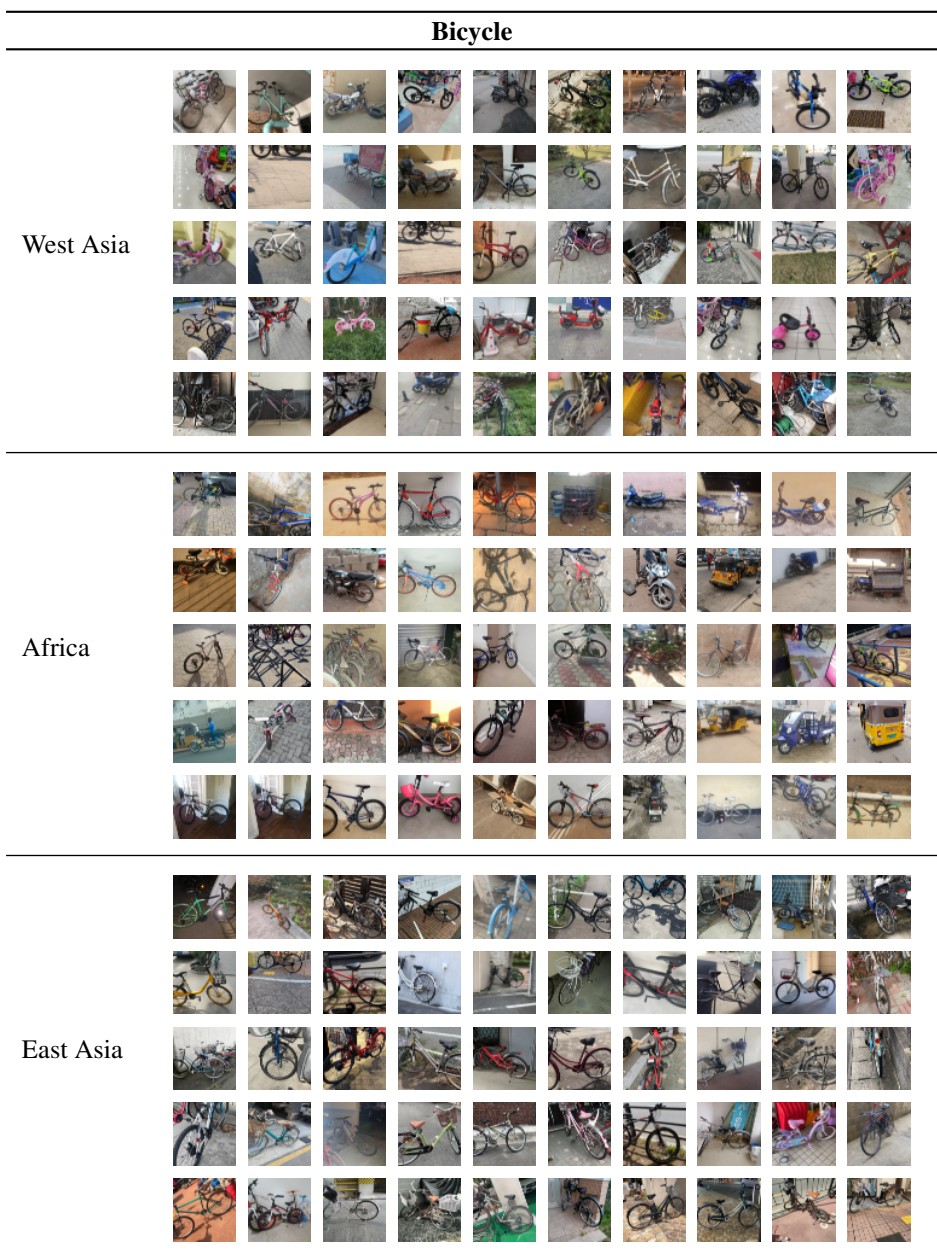

Figure 8: Randomly chosen images for "bicycle" for 3 regions. While most images are of standard bicycles, we notice a couple of interesting images: tricycle (West Asia: r4c9), rickshaws (Africa: r2c8, r2c10, r4c8, r4c10), and motorized cycles (West Asia: r1c8). There are also a lot of children's bicycles.

**Bicycle**

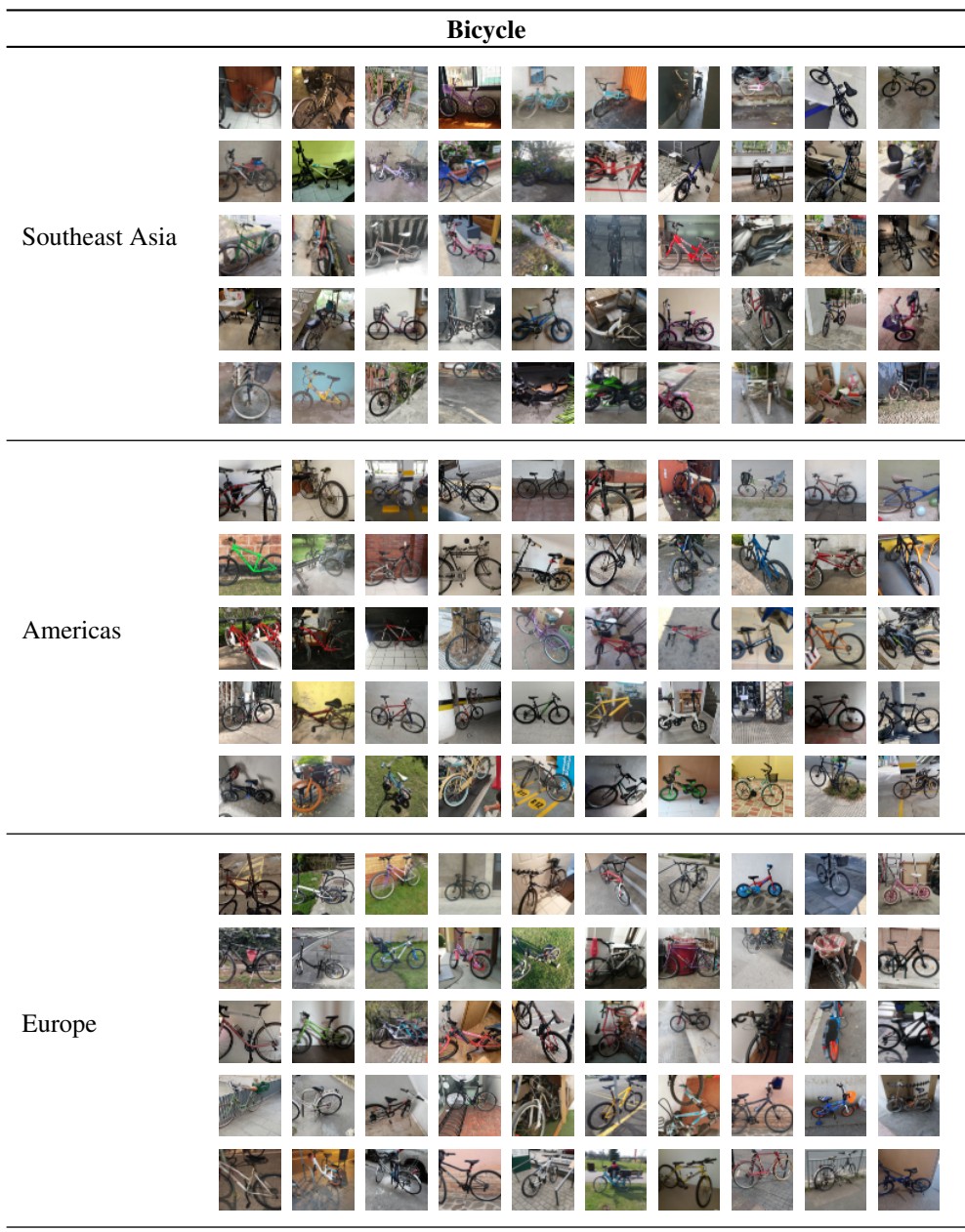

Southeast Asia

Americas

Europe

Figure 9: Randomly chosen images for "bicycle" for the 3 other regions. We see more motorized cycles (Southeast Asia: r5c6) as well as several children's bicycles.

**Boat**

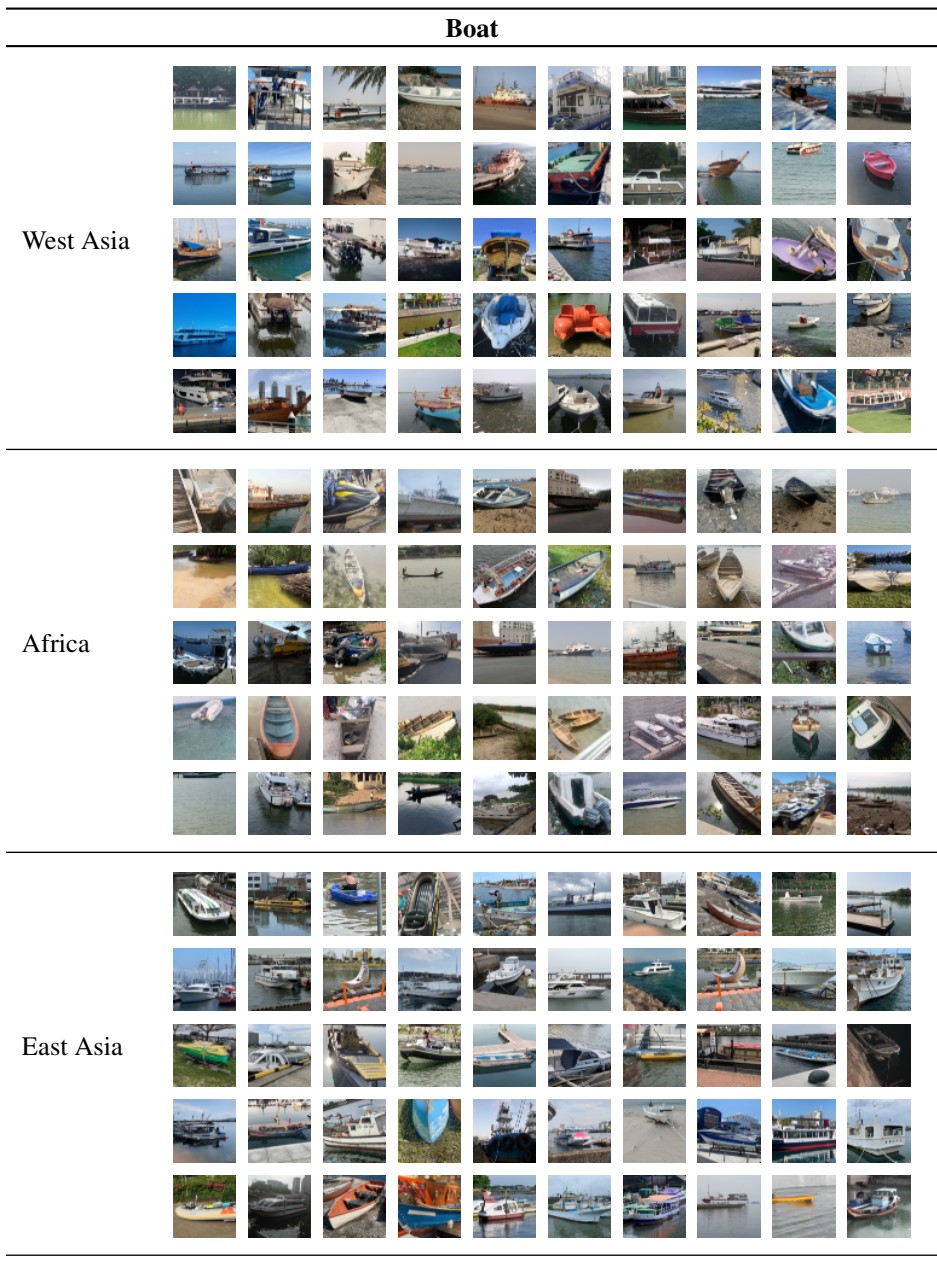

Figure 10: Randomly chosen images for "boat" for 3 regions. We see a variety of boats including larger ships in West Asia (r1c1, r1c2, r1c5, r4c1,r5c1), smaller kayaks and canoes in Africa (r1c8-9, r2c1-4, r4c2 , etc ), and a mix in East Asia.

**Boat**

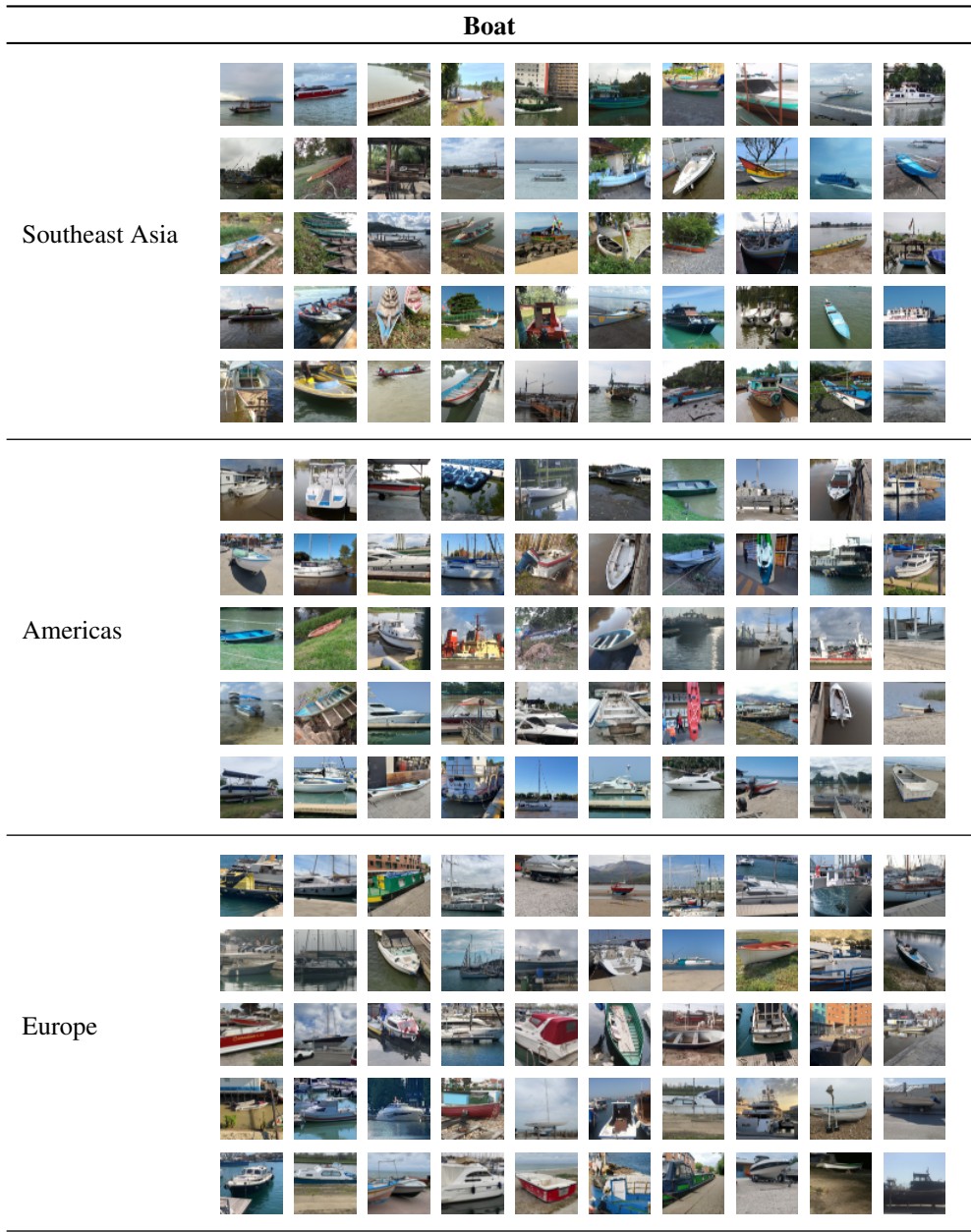

Southeast Asia

Americas

Europe

Figure 11: Randomly chosen images for "boat" for the 3 other regions. We again see a variety of boats ranging from motor boats in Europe and the Americas to smaller boats in Southeast Asia.

**Cleaning equipment**

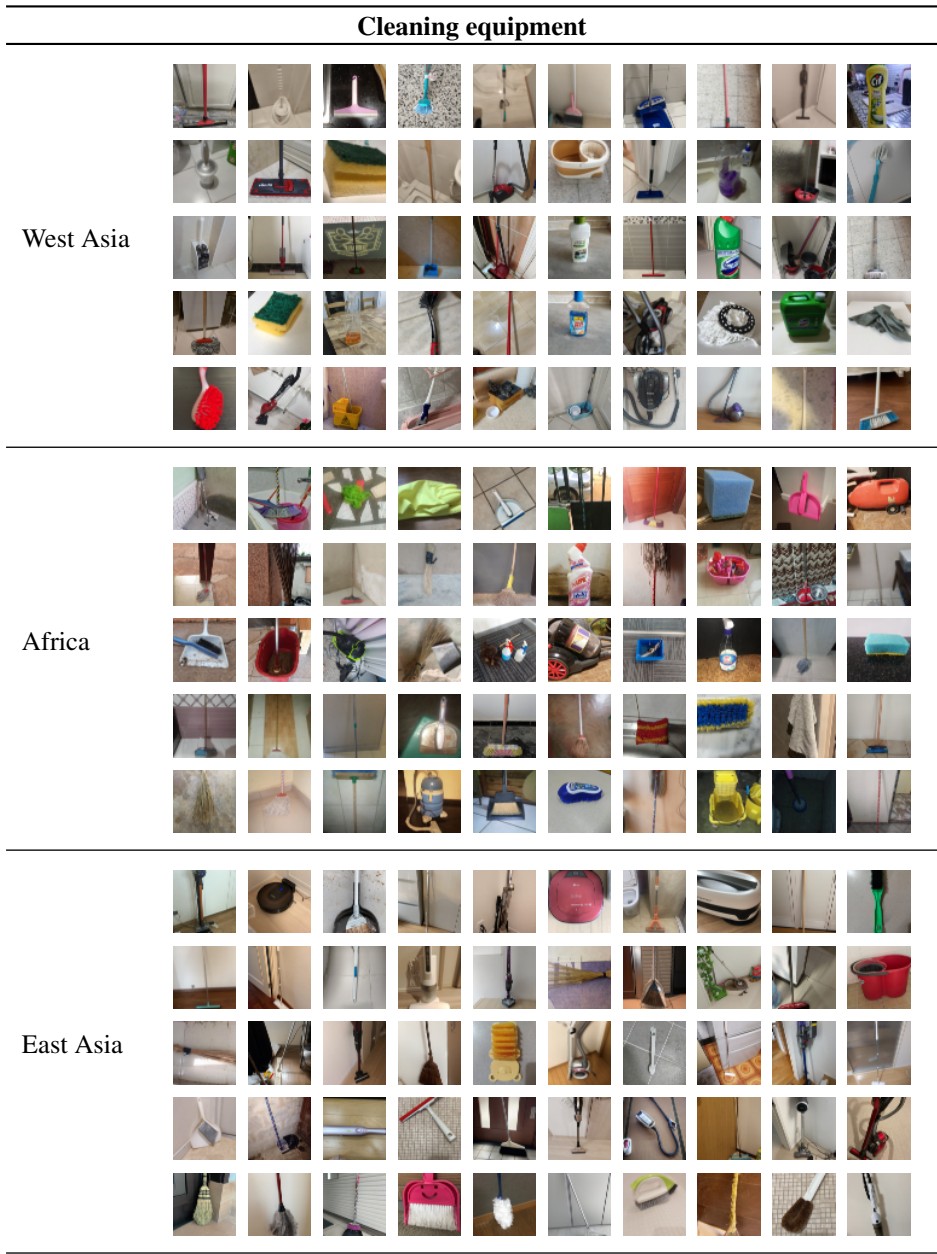

Figure 12: Randomly chosen images for "cleaning equipment" for 3 regions. This appears to be a diverse category within all regions containing images of mops, buckets, products, brooms, etc.

**Cleaning equipment**

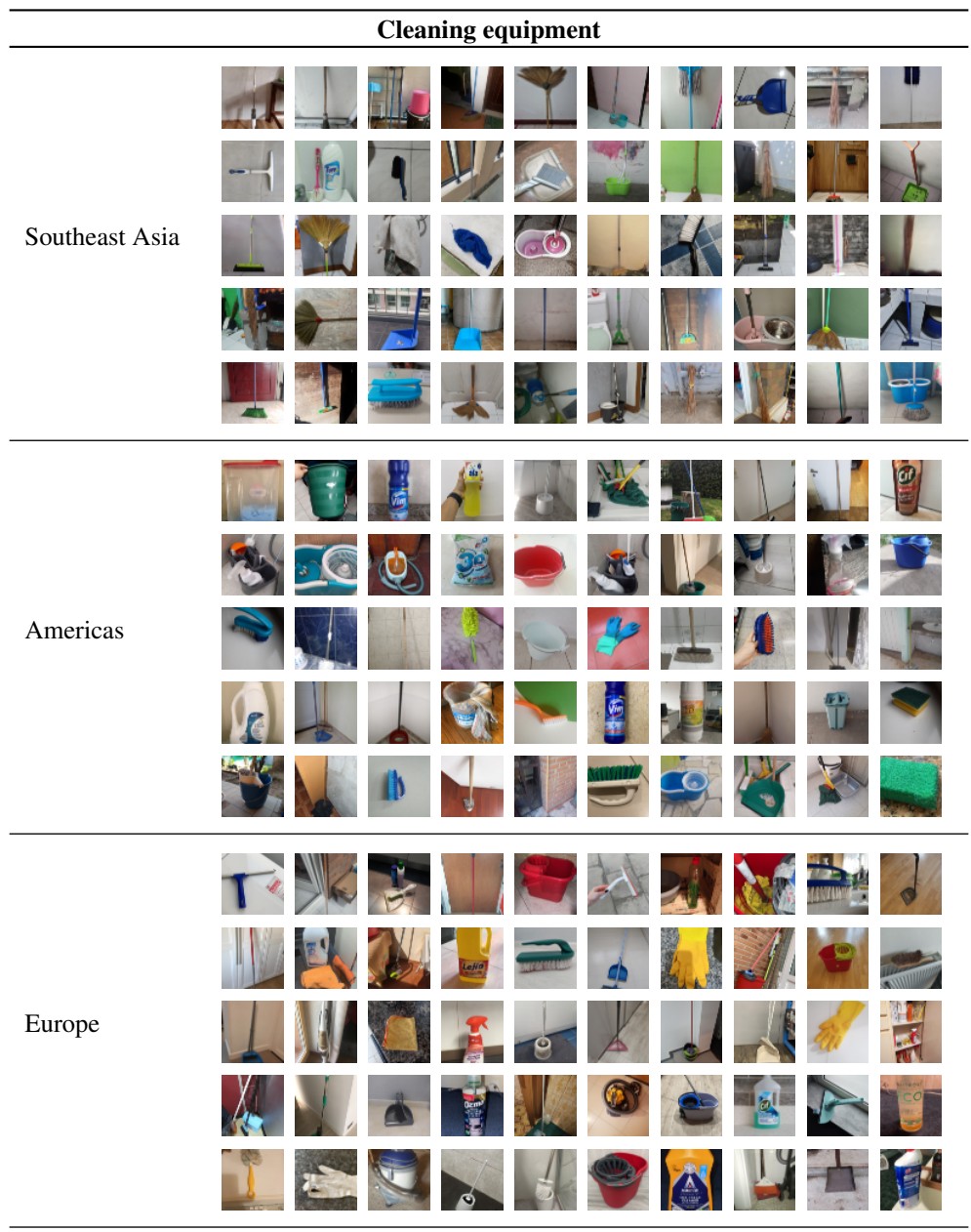

Southeast Asia

Americas

Europe

Figure 13: Randomly chosen images for "cleaning equipment" for the 3 other regions. This appears to be a diverse category within all regions containing images of mops, buckets, products, brooms, etc.

**Spices**

West Asia

Africa

East Asia

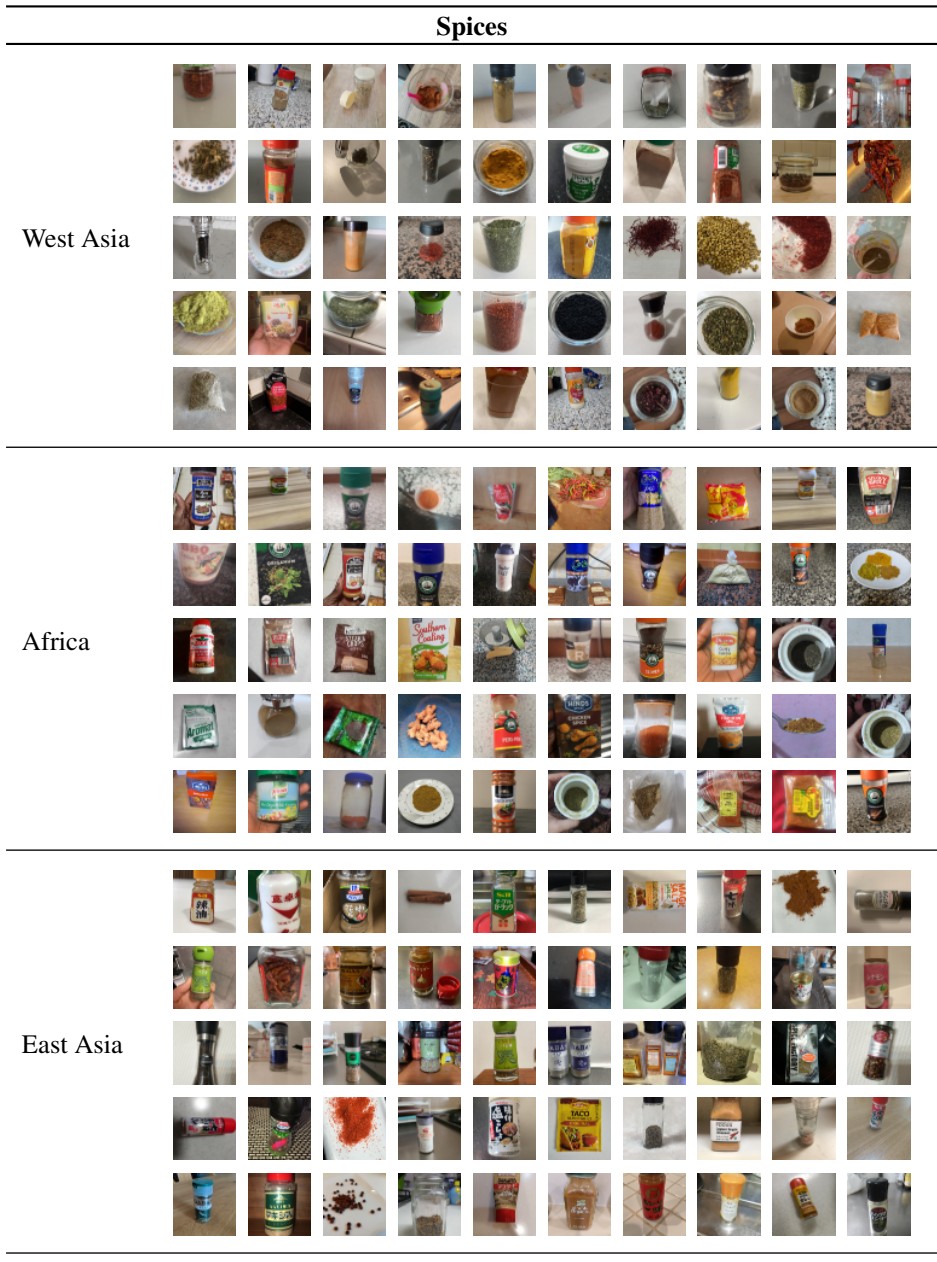

Figure 14: Randomly chosen images for "spices" for 3 regions. We see a wide range of containers, ranging from packets (mostly in Africa), glass jars (in West Asia) to some bottles (all regions).

**Spices**

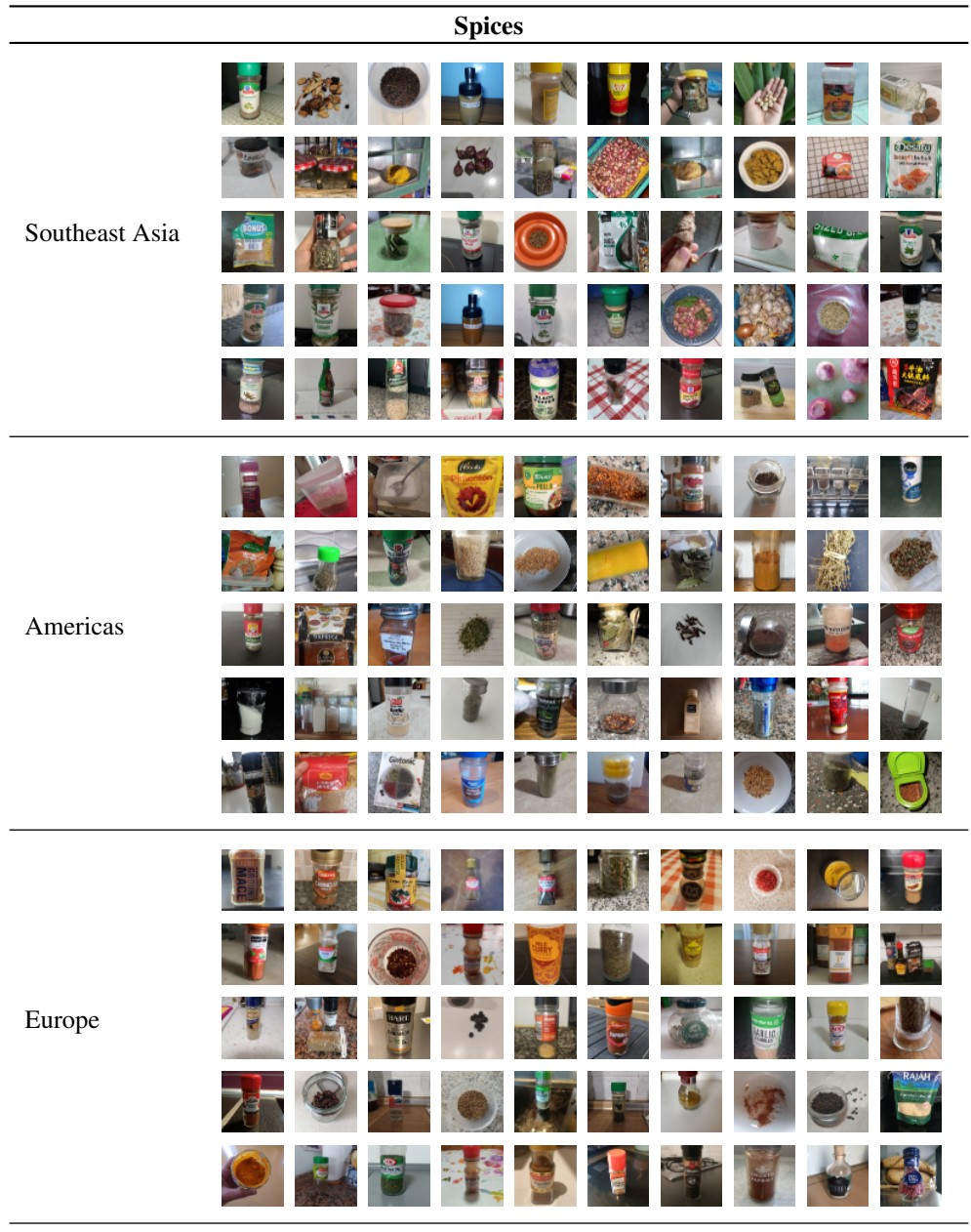

Figure 15: Randomly chosen images for "spices" for 3 regions. We see a wide range of containers, ranging from packets (some in Southeast Asia and Americas) to bottles (some in Southeast Asia)

**Stove**

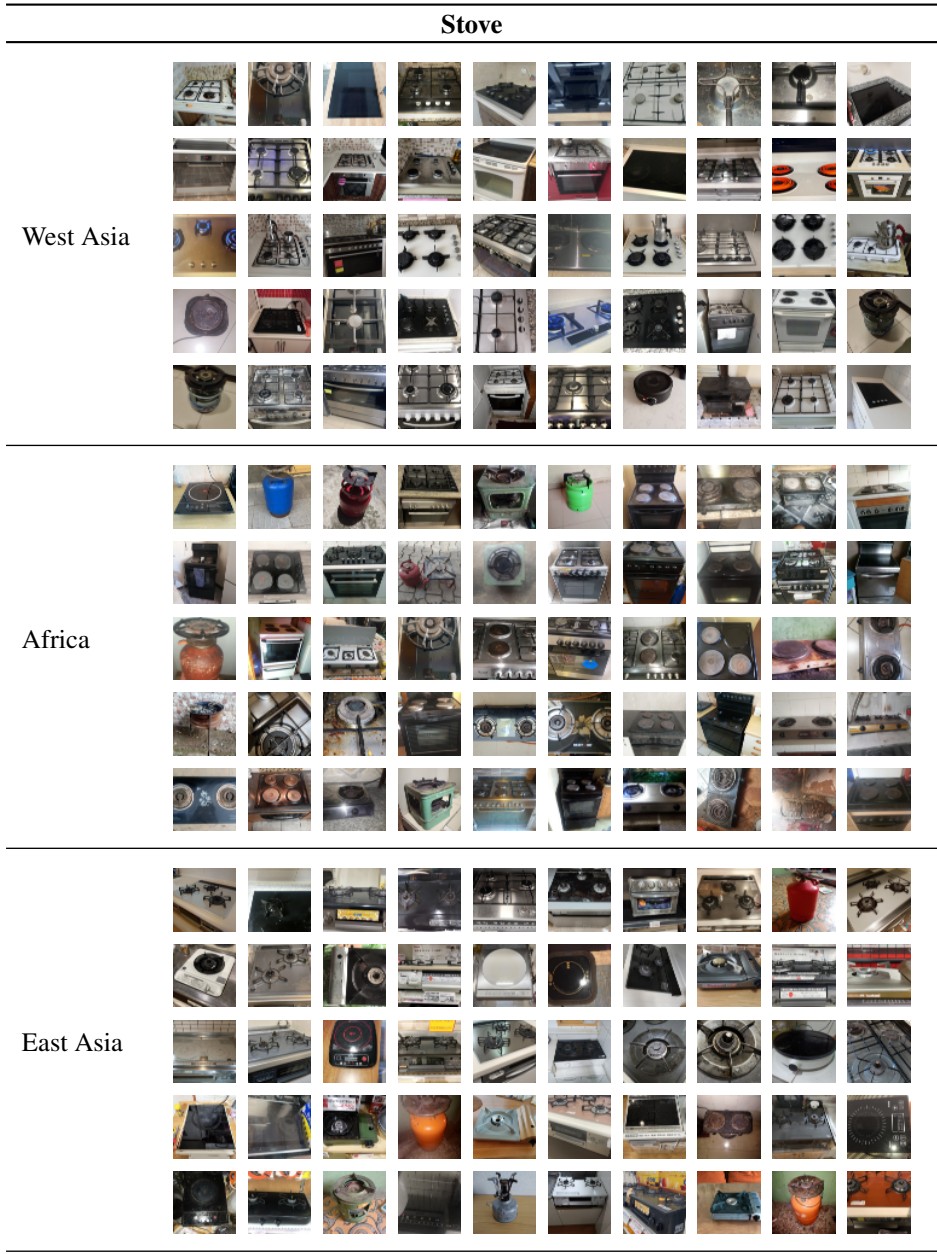

West Asia

Africa

East Asia

Figure 16: Randomly chosen images for "stove" for 3 regions. We see that Africa and East Asia contain one-burner and two burner stoves (along with 4 burner stoves). We also see a variety of stoves in terms of induction, gas, ovens, etc.

**Stove**

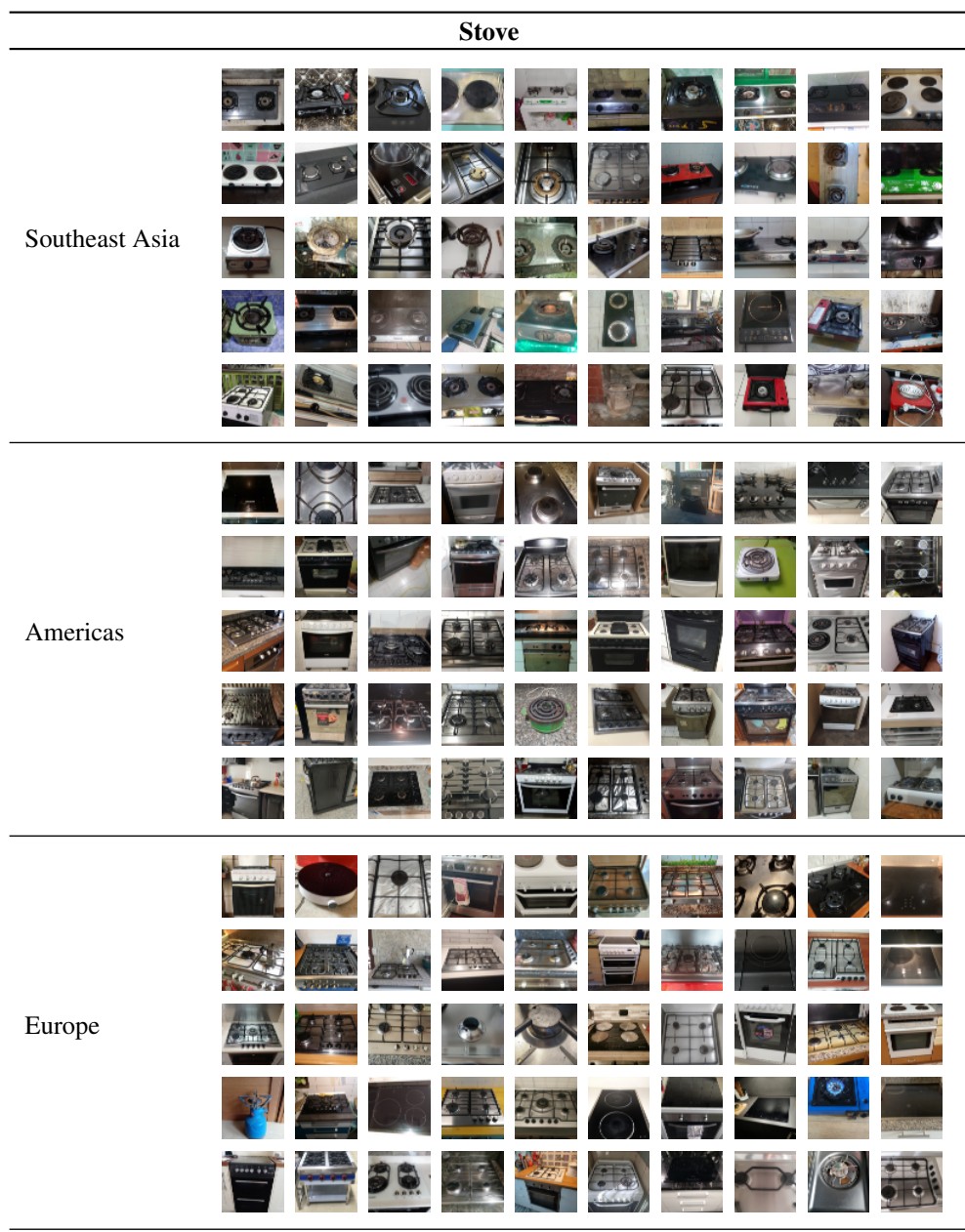

Southeast Asia

Americas

Europe

Figure 17: Randomly chosen images for "stove" for 3 regions. We see that Southeast Asia contains one-burner and two burner stoves (along with 4 burner stoves). We also see a variety of stoves in terms of induction, coils, gas, ovens, etc.

**Waste container**

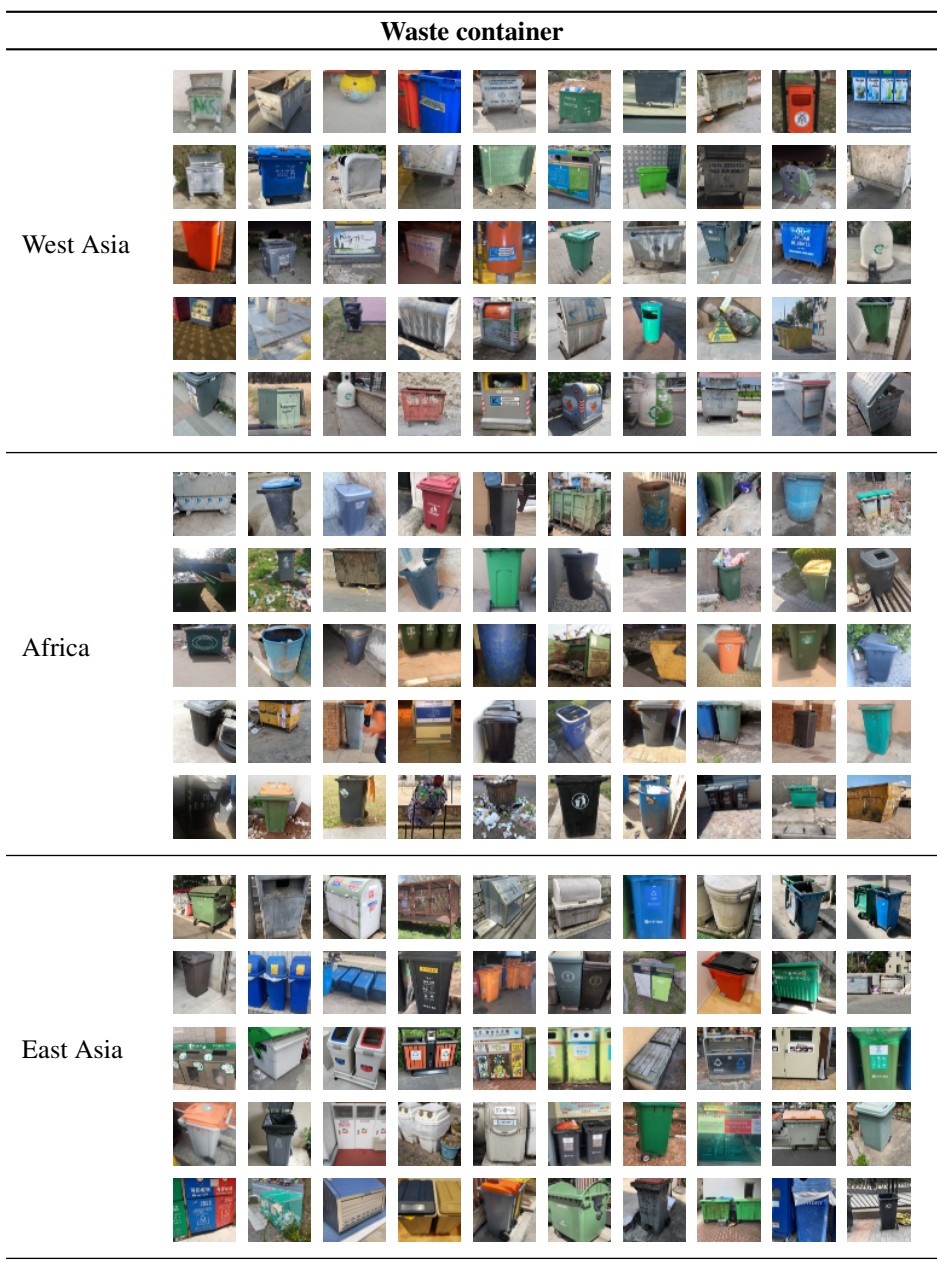

Figure 18: Randomly chosen images for "waste container" for 3 regions. We see that different regions have containers of varying sizes (Africa seems to be smaller than West Asia or East Asia), and have different closing mechanisms (see West Asia r5c6 as an interesting example.) East Asia also tends to have segregated waste containers.

**Waste containers**

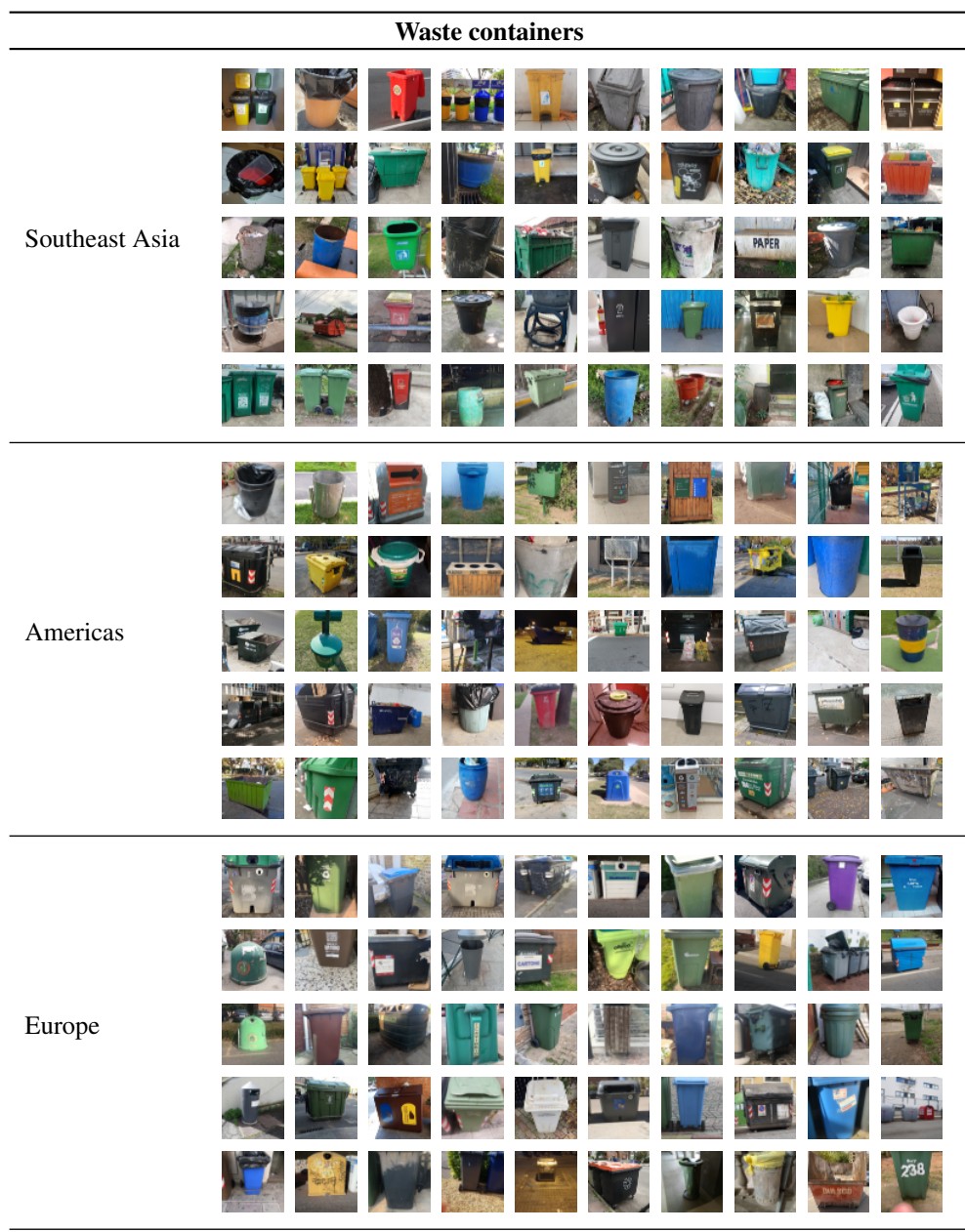

Southeast Asia

Americas

Europe

Figure 19: Randomly chosen images for "waste container" for 3 regions. We see that different regions have containers of varying sizes (Europe seems to have containers of very different sizes) and have different closing mechanisms (see Southeast Asia r2c6 as an interesting example.)

|  | West Asia | Africa | East Asia | Southeast Asia | Americas | Europe |
|---|---|---|---|---|---|---|
| backyard | 216 | 670 | 192 | 218 | 217 | 226 |
| bag | 267 | 397 | 370 | 593 | 298 | 437 |
| bicycle | 237 | 257 | 298 | 235 | 228 | 241 |
| boat | **162** | 227 | **174** | 237 | **84** | 222 |
| bus | 203 | 240 | 223 | 214 | 217 | 226 |
| candle | 232 | 244 | 220 | 239 | 188 | 270 |
| car | 242 | 331 | 276 | 235 | 273 | 363 |
| chair | 279 | 365 | 326 | 512 | 344 | 349 |
| cleaning equipment | 259 | 284 | 307 | 305 | 270 | 361 |
| cooking pot | 216 | 270 | 228 | 202 | 213 | 304 |
| dog | 219 | 194 | 185 | 244 | 206 | 193 |
| dustbin | 220 | 423 | 266 | 203 | 271 | 294 |
| fence | 259 | 322 | 244 | 302 | 226 | 282 |
| flag | 206 | 265 | **139** | 223 | 206 | 272 |
| front_door | 210 | 254 | 216 | 224 | 200 | 235 |
| hairbrush/comb | 269 | 255 | 307 | 300 | 290 | 431 |
| hand soap | 222 | 208 | 277 | 191 | 245 | 362 |
| hat | 209 | 297 | 337 | 316 | 294 | 336 |
| house | 199 | 437 | 208 | 195 | 277 | 194 |
| jug | 217 | 211 | 186 | 249 | 236 | 194 |
| light fixture | 234 | 344 | 248 | 209 | 191 | 300 |
| light switch | 215 | 240 | 246 | 273 | 273 | 234 |
| lighter | 221 | 312 | 225 | 237 | 217 | 268 |
| medicine | 242 | 286 | 310 | 330 | 328 | 300 |
| monument | **161** | 191 | 186 | 183 | 254 | 245 |
| plate of food | 211 | 480 | 294 | 364 | 241 | 304 |
| religious building | 222 | 230 | 204 | 226 | 197 | 229 |
| road sign | 226 | 416 | 258 | 270 | 235 | 284 |
| spices | 243 | 250 | 331 | 216 | 290 | 300 |
| stall | **143** | 215 | 203 | 227 | 197 | 221 |
| storefront | 209 | 306 | 191 | 240 | 243 | 204 |
| stove | 199 | 553 | 191 | 262 | 206 | 282 |
| streetlight / lantern | 202 | 346 | 211 | 196 | 208 | 227 |
| toothbrush | 264 | 258 | 330 | 361 | 337 | 270 |
| toothpaste / toothpowder | 209 | 288 | 269 | 230 | 245 | 315 |
| toy | 224 | 221 | 280 | 292 | 323 | 287 |
| tree | 226 | 308 | 245 | 357 | 300 | 328 |
| truck | 205 | 246 | 207 | 231 | 212 | 225 |
| waste container | 231 | 213 | 209 | 213 | 211 | 253 |
| wheelbarrow | **122** | 267 | **130** | 197 | **152** | 243 |

Table 3: We show the counts of objects per region in GeoDE. **Bolded** are the ones categories for which we were not able to get 175 images per region.

[8] William A Gaviria Rojas, Sudnya Diamos, Keertan Ranjan Kini, David Kanter, Vijay Janapa Reddi, and Cody Coleman. The dollar street dataset: Images representing the geographic and socioeconomic diversity of the world. In *NeurIPS Datasets&Benchmarks Track*, 2022.

[9] Emma Strubell, Ananya Ganesh, and Andrew McCallum. Energy and policy considerations for deep learning in nlp. In *ACL*, 2019.

[10] Antonio Torralba and Alexei A Efros. Unbiased look at dataset bias. In *CVPR*, 2011.