# OpenReview forum: "GeoDE: a Geographically Diverse Evaluation Dataset for Object Recognition"
_NeurIPS.cc/2023/Track/Datasets_and_Benchmarks — NeurIPS 2023 Datasets and Benchmarks Poster_

### Official Review · Reviewer_8Bhq · 2023-07-21
**Dataset for advancing Geographically Diverse Object Recognition**

**Rating:** 7
**Confidence:** 3

**Strengths:**

The paper showcases several strengths, outlined as follows:
* The paper presents a meticulous approach to collecting images from diverse regions, thoroughly addressing privacy, consent, and copyright considerations.
* The paper introduced the GeoDE dataset, a novel labeled image dataset specifically tailored for geographically diverse object recognition. This dataset surpasses the DollarStreet dataset regarding the number of included images, enhancing the potential for more comprehensive and accurate model training and evaluation.
* The paper uncovers regional variations in the appearance of objects, demonstrating how these differences directly impact the accuracy of current object recognition methods.

**Additional Feedback:**

Please refer to the Opportunities For Improvement section for comments and suggestions.

**Clarity:**

The paper is well-written, and the theme of building diverse image datasets worldwide while considering ethics and privacy is well-explained.

**Correctness:**

The paper's claims appear correct, but a more detailed country-by-country analysis is needed.
The dataset construction and evaluation are sound.

**Documentation:**

The dataset will be made publicly available, and the evaluation process is clear and seems to be reproducible.

**Ethics:**

Ethics and privacy were taken into consideration in the creation of this dataset, and the process is clearly explained in the paper.

**Limitations:**

The author adequately addresses the negative societal impact.

**Opportunities For Improvement:**

The weaknesses of the paper are outlined as follows:
* The paper pointed out that the data collection method is not easily scalable due to its high cost. This limitation hinders the potential for obtaining a large amount of data, which could impact the dataset's representativeness and diversity.
* The paper's division of geography is too coarse. The broad geographical division may not adequately capture the differences in object appearances among countries. For example, even in East Asia, objects look pretty different in India and Japan (and even China and Japan). This lack of granularity can introduce unknown data biases. Country-by-country comparisons and analysis are also needed.
* Despite contributing to the development of data collection methodologies, the paper's GeoDE dataset has limited differentiation compared to the DollarStreet dataset.

**Relation To Prior Work:**

This paper presents relevant studies and conducts fair comparisons.

**Summary And Contributions:**

This paper presents the GeoDE dataset, designed for training and evaluating object recognition methods while considering the geographical diversity of common objects worldwide.
The dataset offers several advantages compared to existing datasets:

* It addresses copyright and privacy concerns related to identifiable individuals by employing a meticulous data collection approach, setting it apart from web-scraped image datasets like GeoYFCC.

* The GeoDE dataset is unique because it does not overlap with the existing geo-aware image dataset DollarStreet. Moreover, it successfully compiles numerous images specific to six distinct global regions from scratch.

Furthermore, the authors' experiments shed light on the challenges posed by the diverse appearance of objects in different global regions, making object recognition a complex task. The authors demonstrate that using GeoDE as training data can mitigate this problem effectively.

---

> ### Author Response · Authors · 2023-08-15
> **Response to 8Bhq**
>
> We thank the reviewer for their thorough review. We address the scalability and comparison to DollarStreet in the overall response, and address other concerns here.
>
> **Region size.** We agree with the reviewer that the categorization of the world into 6 large geographic regions is quite broad. This scale was chosen to be small enough to capture geographic differences, while still being cost-effective. Our intuition was that many objects would be expected to look similar within different countries of the same region, due to likely economic and social ties.  To validate this intuition, we run an additional set of experiments (Appendix E.3). We compute accuracies for individual countries for a CLIP model as well as  when training a linear model on GeoDE and ImageNet. When using a pre-trained model like CLIP, we do notice differences between accuracies of countries within each region, however, we note that the overall trend in accuracy remains roughly the same. When training with images from GeoDE, discrepancies between countries within each region further reduces.

---

> > ### Comment · Reviewer_8Bhq · 2023-08-23
> > **Thanks for your response**
> >
> > Thanks for the follow-up work. My concerns have been almost addressed. I would like to increase my score to 7.

---

### Official Review · Reviewer_1Lpg · 2023-07-21
**Cool new geographically diverse visual dataset**

**Rating:** 6
**Confidence:** 4
**Correctness:** Seems correct.
**Clarity:** yes.

**Strengths:**

Great work on building a dataset. Careful attention to licenses. No PII. Good training results. Generally I like it.

**Additional Feedback:**

"the question remains whether a geographically diverse dataset can be constructed from scratch."

I don't think anyone questions if this is possible. It's obviously possible and seems like a false strawperson. Honestly I think the paper would be strongly without obviously false rheotorical flourishes.

I increased my rating to 6. I think 'object centric' vs. 'action centric' is a false dichotomy without any additional proof support. Good work to the team!

**Documentation:**

I don't see any long term data maintenance plan. That is an opportunity for improvement.

**Ethics:**

Ethically looks good.

**Limitations:**

It's a modest improvement over existing approaches. But it's also expensive and not really scalable.

I did not review the images for general quality characteristics.

**Opportunities For Improvement:**

What would you say is the biggest improvement over dollar street?

Sounds to me like (1) size, (2) better geo diversity, (3) no PII, (4) better object distribution. Is that right?

What resolutions are the photos? Min size was discussed, but no real distributional analysis or comparison against existing datasets.

Is there any way to compare against training (or retraining) with Dollar street (or combining the two in some way)? If you can answer this question, I will improve my rating!

**Relation To Prior Work:**

Yes

**Summary And Contributions:**

Manually collected, diverse dataset for visual recognition. Reasonably sized and no PII. Improves upon best in class existing datasets (namely Dollar Street) in some ways. Good attention to geographic balance.

---

> ### Author Response · Authors · 2023-08-15
> **Response to 1Lpg**
>
> We thank the reviewer for their insightful questions and comments! We address the comparison to Dollar Street and scalability of the dataset in the overall response, and address individual concerns here.
>
> **Long term data maintenance.** We’ve updated the datasheet in the appendix to reflect this (lns 267-271). Currently, the dataset is being hosted at https://geodiverse-data-collection.cs.princeton.edu/. For a more long term solution, we’re considering one of two options: either partnering with the Computer Vision Data Foundation  (CVDF; http://www.cvdfoundation.org/) or utilizing https://researchdata.princeton.edu/news/2023-05-25/coming-soon-princeton-data-commons
>
> **Image resolutions.** The median GeoDE image is 1080 x 1440 pixels, whereas the median ImageNet image is 375x500 pixels.
>
> **Additional feedback.** Thank you for the feedback, we’ve rewritten this line, iterating that we take an object-centric collection approach. (lns 35-37).

---

> > ### Comment · Reviewer_1Lpg · 2023-08-19
> > **Object centric collection?**
> >
> > Can you help me understand the meaningful benefits of an object-centric collection vis a vis DollarStreet? That is one of the claimed differentiations, and I'm not sure I understand the real benefit.
> >
> > Thank you for the resolution information and hosting plans!

---

> > > ### Author Response · Authors · 2023-08-21
> > >
> > > We would like to clarify that there isn't an inherent benefit for an object-centric collection vs. action centric. These 2 approaches lead to datasets that allow us to understand different things about the world. For example, GeoDE's collection approach would let us understand how (e.g), beds look across the globe, whereas DollarStreet's collection approach (since they use labels of "guest bedroom", vs. "bedroom") would let us understand how hosting customs differ around the world.

---

### Official Review · Reviewer_TgPt · 2023-07-25
**The review mainly focuses on the rationality and necessity of the paper.**

**Rating:** 7
**Confidence:** 4
**Correctness:** correct
**Clarity:** Written is clear.

**Strengths:**

The authors collected a geographically diverse dataset of common objects to provide the diversified model evaluation.

GeoDE enables control of the image distribution to reduce dataset biases and the data is an unseen test set at least temporarily.

GeoDE dataset benefits in remedying some of the concerns with large-scale web-scraped datasets.

The dataset is selected in six regions, which has more universal.

Written is clear.


**Additional Feedback:**

Please see the above reviews.

**Documentation:**

enough

**Ethics:**

There are no or only very minor ethics concerns.

**Limitations:**

I think this paper still needs to supplement the comparison of datasets.

**Opportunities For Improvement:**

GeoDE is an object recognition dataset, I suggest that this dataset needs to be compared with a large object detection dataset like COCO, which contains 80 categories with a large number of images. Because I think detection is a more complex recognition task.




**Relation To Prior Work:**

I think this paper still needs to supplement the comparison of datasets.

**Summary And Contributions:**

This paper introduces GeoDE, a geographically diverse dataset with 61,940 images from 40 classes and 6 world regions, and no personally identifiable information, collected by soliciting images from people across the world. This is a more universal dataset.

---

> ### Author Response · Authors · 2023-08-15
> **Response to TgPt**
>
> We thank the reviewer for their positive review, and praising the dataset itself. We note that GeoDE is an object recognition dataset, with each image containing only object names but no bounding boxes around them. Thus, our experiments involved fine-tuning on the object-centric ImageNet dataset. Comparing GeoDE to scene-centric object detection datasets like COCO or Open Images would have to be done through some version of cropping the scene images around the target objects, which may introduce its own biases into the analysis.

---

### Author Response · Authors · 2023-08-15
**Overall response**

We thank the reviewers for their comments and suggestions, and praising the geodiversity of the dataset (all) as well as the collection method, which takes into account privacy concerns (1lpg, 8bhq). We address some common concerns here.

**Scalability of this collection method (1lpg, 8bhq).** We agree that a key disadvantage of this collection method is the cost. However, as we show in Sections 6 and 7, even with the limited size, we are able to identify biases in current models (e.g., Fig. 8) as well as improve current models through fine tuning (e.g., Tab. 5). Moreover, as pointed out in lines 38-41, with the increase in multimodal foundational models, we might need to identify different methods of collecting data to ensure that we do not violate the core ML tenet of separate train/test splits.

**Comparison with DollarStreet (1lpg, 8bhq).** We have updated our paper to include a more detailed comparison with DollarStreet (lns 106-116 in the main paper, lns 348-365 in the supplementary material). We summarize the discussion here. Since DollarStreet was adapted from an existing collection of images taken by Gapminder taken to show how people across the world live, this dataset is collected through querying for actions rather than objects. Thus, the DollarStreet dataset  allows us to understand how different objects are used for similar tasks in different parts of the world. One such example is in the category of “beds”, where photos are taken of main bedrooms, guest bedrooms, and kids’ bedrooms. While these classifications are extremely interesting from a social science lens, they are less interesting from an object recognition lens. Thus, while constructing our dataset, we take an object-centric approach, asking participants to take photos of various objects around them.

We compare these datasets in the appendix C.2 to understand the value of an image, and find that in general, training on DollarStreet improves performance on GeoDE and vice versa, however, (as expected), each performs better on in-distribution data.

---

### Decision · Program_Chairs · 2023-09-22

**Decision:**

Accept (Poster)

**Comment:**

All the reviewers appreciated the proposed dataset of geographically diverse objects, which may reduce dataset biases in object recognition and remedy generalization issues with large-scale web-scraped datasets. The authors' rebuttal also addressed most of the main concerns of the reviewers and AC find no critical issue, thus recommending acceptance.